# An ontogeny-cytokine code determines macrophage response polarity and tumor outcomes
Dominik J. Schaer, Nadja Schulthess-Lutz ⓘ , Matthias J. Peterhans ⓘ , Livio Baselgia ⓘ ,
Melanie Eschment ⓘ , Rok Humar & Florence Vallelian ⓘ ✉

Tumor-associated macrophages can either promote or suppress cancer, but therapeutic targeting remains challenging because we lack a predictive framework for macrophage function. The prevailing M1/M2 paradigm oversimplifies how macrophage developmental origin (ontogeny) and local cytokines shape antitumor versus protumor behavior. We systematically map eight reference macrophage states by differentiating mouse bone marrow cells with M-CSF or GM-CSF and polarizing them with four key cytokines (IFN-γ, IL-4, IL-10, TGF-β). Using integrated transcriptomic profiling, 3D tumor spheroids, and experimental metastasis models, we find that macrophage ontogeny determines whether cytokines promote or suppress tumor progression. Most notably, IL-4 induces opposite effects depending on ontogeny: promoting tumor growth, invasion, and metastasis in M-CSF-derived macrophages, while suppressing these processes in GM-CSF-derived macrophages. A similar ontogeny-dependent divergence was observed for IL-10, whereas IFN-γ consistently exerted antitumor effects and TGF-β protumor effects across both lineages. These findings define an ontogeny-cytokine interaction framework that determines macrophage function based on developmental origin and cytokine context. By identifying ontogeny as a key determinant of cytokine responses, this work provides a conceptual basis for more precise macrophage-directed cancer immunotherapy strategies.

Macrophages dominate the immune landscape of solid tumors and can either restrain or accelerate malignancy[1]. Depletion or reprogramming of tumor-associated macrophages (TAMs) with CSF1R inhibitors, TLR, STING, or CD40 agonists, or interferon-γ (IFN-γ) slows tumor growth in mice, yet durable clinical responses remain rare[2–5]. Thus, we lack a comparative, multi-scale framework for how TAMs integrate developmental and environmental signals to adopt pro- versus antitumor functions[6–8].

TAM heterogeneity reflects two critical but poorly integrated factors[9–11]. First, macrophage ontogeny: a homeostatic lineage maintained by macrophage colony-stimulating factor (M-CSF) coexists with an inflammatory lineage expanded under granulocyte–macrophage colony-stimulating factor (GM-CSF)[12,13]. Second, local cytokine milieu: soluble mediators such as IFN-γ, interleukin-4 (IL-4), interleukin-10 (IL-10), and transforming growth factor-β (TGF-β) drive macrophages along an activation spectrum historically summarized as "M1/M2"[14]. Whether and how these two axes—ontogeny and cytokine environment—interact to determine TAM function remains unclear, hampering rational therapeutic design[9–11].

Here, we introduce a reductionist yet physiologically informative pipeline that systematically dissects the interplay between ontogeny and polarization signals. Bone marrow (BM) precursors were differentiated with M-CSF or GM-CSF and subsequently polarized with IFN-γ, IL-4, IL-10, or TGF-β, generating eight precisely defined macrophage states.

We integrated bulk and single-cell RNA sequencing with high-content imaging, 3D tumor spheroid co-cultures, invasion assays, and an experimental metastasis model to link each macrophage state with distinct transcriptional signatures, effects on cancer cell proliferation and invasion, and capacity to modulate pulmonary metastatic seeding in vivo. This systematic approach reveals how developmental origin constrains or amplifies the functional consequences of environmental polarization, providing a quantitative framework for TAM-directed therapies.

Department of Internal Medicine, University Hospital and University of Zurich, Zurich, Switzerland. ✉ e-mail: florence.vallelian@usz.ch

## Results

### Macrophage ontogeny and cytokine environment create distinct transcriptional landscapes

We generated eight macrophage states using our cytokine-ontogeny matrix (Fig. 1A), enabling direct comparison of how identical cytokine signals affect macrophages with different developmental origins. Bulk RNA-sequencing (RNA-seq) was performed on five biological replicates per condition. Changes in transcriptional programs were visualized by principal component analysis (PCA), followed by gene-set enrichment analysis (GSEA) of hallmark gene sets and transcription factors (TFs). In a PCA across all eight conditions (Fig. 1B), PC1 (54% variance) segregated macrophages primarily by ontogeny: GM-CSF states loaded negatively, while M-CSF-derived states loaded positively (Fig. 1B). GSEA of genes contributing to negative PC1 loadings highlighted NF-kB-regulated inflammatory programs, consistent with the intrinsic inflammatory bias of GM-CSF macrophages[15,16]. This ontogeny-driven separation was so pronounced that it exceeded the variance explained by individual cytokine treatments, indicating that developmental origin fundamentally shapes macrophage identity (Fig. 1B).

The ontogeny-cytokine interaction was most evident when each lineage was examined separately. Within GM-CSF macrophages (Fig. 1C, D), IL-4 again separated along PC1 (47% variance) but showed no additional significant Hallmark enrichment beyond NF-kB trajectory. PC2 (25% variance) distinguished the IFN-γ pole characterized by IRF1/STAT1-regulated genes (*Gbp* genes, *Cxcl9*), from IL-4 and IL-10/TGF-β poles, denoting distinct remodeling programs. In contrast, M-CSF macrophages displayed a different response landscape (Fig. 1E, F). IL-4 treatment again dominated PC1 (54% variance), but here GSEA revealed robust enrichment of reparative programs (EMT, UPR) with concurrent repression of interferon-response hallmarks. TF analysis aligned with this shift—MYC, LEF1, CTNNB1, ESR1, TWIST2 and SMAD3 increased—while IRF family members were comparatively depleted. Along PC2 (26% variance), M-CSF macrophages spanned an inflammatory-to-remodeling continuum: IFN-γ-treated cells anchored one pole with the chemokine-rich, IFN-responsive signatures (*Ccl7, Ccl2, Ccl12*) (GSEA: IFN-γ; TFs: STAT1/IRF1), whereas TGF-β-treated cells anchored the opposite pole with matrix-remodeling signatures (*Mmp14, Pmepa1, Notch4*) (GSEA: TGF-β signaling, EMT; TFs: SMAD1/4); IL-10 occupied an intermediate position along PC2 between the IFN-γ (chemokine-rich) and TGF-β (matrix-remodeling) poles. These data establish that identical cytokine treatments produce lineage-dependent transcriptional programs, providing the transcriptional basis for functional differences.

### Single-cell RNA-sequencing resolves IL-4 and IFN-γ into distinct inflammatory trajectories within GM-CSF macrophages

Compared to M-CSF macrophages, GM-CSF macrophage cultures are intrinsically heterogeneous, generating both macrophages and a distinct population of MHC-II+ Ccl22+ Ccr7+ dendritic cell-like inflammatory macrophages[17,18]. To determine whether the transcriptional axes identified by bulk RNA-seq reflect uniform population-wide shifts or arise from rare subpopulations within GM-CSF cultures, we performed multiplexed scRNA-seq on 49,441 GM-CSF–derived macrophages across all cytokine treatment conditions. Uniform Manifold Approximation and Projection (UMAP) of the combined dataset revealed treatment-specific clustering patterns (Fig. 2A). Single-cell PCA (Fig. 2B; PC1 = 13.21%, PC2 = 4.65% variance) recapitulated the bulk RNA-seq patterns: IL-4 treatment displaced macrophages farthest along PC1 towards a cell cluster enriched for NF-kB-controlled gene expression (Fig. 2C) with high *Ccl22* and *Ccr7* (Fig. 2D), consistent with a dendritic-cell-like, T-cell-activating phenotype. This contrasts with the canonical anti-inflammatory expectation for IL-4 and supports our hypothesis that ontogeny fundamentally alters cytokine signaling outcomes.

In contrast, IFN-γ-treated cells loaded strongly along the positive axis of PC2, maintaining high MHC-II expression (Fig. 2B) while acquiring a complement/interferon gene module (*C1qb, C1qc, Gbp2, Irf1*, Supplementary Fig. 1); transcription factor enrichment analysis highlighted IRF8, IRF1, and STAT1, confirming an interferon-responsive signature (Fig. 2C).

TGF-β and IL-10 macrophages colocalized in the lower-left quadrant (negative for both PC1 and PC2), where they co-expressed reparative markers including *Chil3* and *Cd24* (Fig. 2D), forming an immunosuppressive, trophic branch distinct from either inflammatory pole. TGF-β-treated cells clustered more distinctly toward the negative PC2 pole, while IL-10-treated macrophages occupied an intermediate position along PC2, between the IFN-γ (positive) and TGF-β (negative) extremes.

To validate these single-cell expression patterns, we examined key marker genes across treatment conditions using violin plots (Supplementary Fig. 2). The top PC1 loading genes—*Ccl22* and *Ccr7* were highly expressed in IL-4-treated cells. Similarly, the top PC2 loading genes— *Ly6a* and *C1qb* clearly distinguished IFN-γ-treated cells from IL-4-treated cells. These single-cell expression signatures were independently confirmed by our bulk RNA-seq data (Supplementary Fig. 2).

Thus, within the GM-CSF macrophage lineage, IL-4 and IFN-γ drive orthogonal inflammatory trajectories, whereas TGF-β consolidates a shared remodeling program. To test whether these transcriptional differences translate into functional outcomes, we next examined protein expression and cellular behaviors.

### Ontogeny-cytokine interactions control the expression of key immunosuppressive markers

To connect transcriptional states (Fig. 3A) with functional phenotypes, we examined protein expression of ARG1 and SPP1, hallmark markers of immunosuppressive TAMs[19,20]. We tracked fluorescence over 100 h using live cell microscopy (Incucyte) of M- and GM-CSF macrophages from Arg1-YFP and Spp1-IRES-tdTomato reporter mice (Fig. 3B, time-course representative of three independent experiments). IL-4-treatment selectively induced ARG1 fluorescence in M-CSF macrophages, which adopted elongated, high-eccentricity morphologies (Fig. 3B, C). In contrast, IL-10— and, to a lesser extent, TGF-β— induced SPP1 selectively in GM-CSF macrophages. This aligns with prior work showing that GM-CSF–driven macrophage differentiation promotes SPP1 linked to aggressive tumor behavior in lung adenocarcinoma[21]. These cells maintained a compact, low-area morphology (Fig. 3C, D). Control and IFN-γ-stimulated macrophages kept the expression of both ARG1 and SPP1 at low levels.

Quantitative morphological analysis revealed ontogeny-specific signatures. GM-CSF macrophages displayed a spectrum of predominantly round cells with variable cell areas. The SPP1+ IL-10– and TGF-β–polarized GM-CSF macrophages exhibited low area morphology, while the other states were represented by larger cells. M-CSF macrophages were discriminated by morphological eccentricity. IL-4– and TGF-β–polarized cells acquired highly elongated shapes, while unpolarized and IFN-γ-treated cells remained more rounded (Fig. 3C, D).

To assess functional consequences, we examined antigen presentation capacity. M-CSF macrophages uniformly lacked surface MHC-II and failed to stimulate OT-II CD4 T cell proliferation[18]. In contrast, GM-CSF macrophages produced divergent cytokine-dependent outcomes (Fig. 3E, F): IFN-γ and IL-4 preserved high MHC-II levels and elicited OT-II-specific T-cell division, whereas IL-10 or TGF-β polarized macrophages downregulated MHC-II and lost T-cell activation function.

Together, these data support our transcriptional findings at the protein and functional levels, establishing a CSF-cytokine signaling gate that generates distinct immunosuppressive macrophage states with protumor potential: ARG1-high, IL-4-driven states arise exclusively in the M-CSF lineage and are characterized by elongated morphology and absent antigen presentation, whereas SPP1-high, IL-10/TGF-β-driven states arise in the GM-CSF lineage and display compact morphology with selective loss of T-cell stimulatory capacity.

### Macrophage phenotype determines cancer cell fate in 3D co-cultures

To determine how M-CSF macrophage states influence cancer cells, we conducted time-resolved, multiplexed scRNA-seq of mixed 3D spheroids containing M-CSF macrophages and MC38 colon carcinoma cancer cells

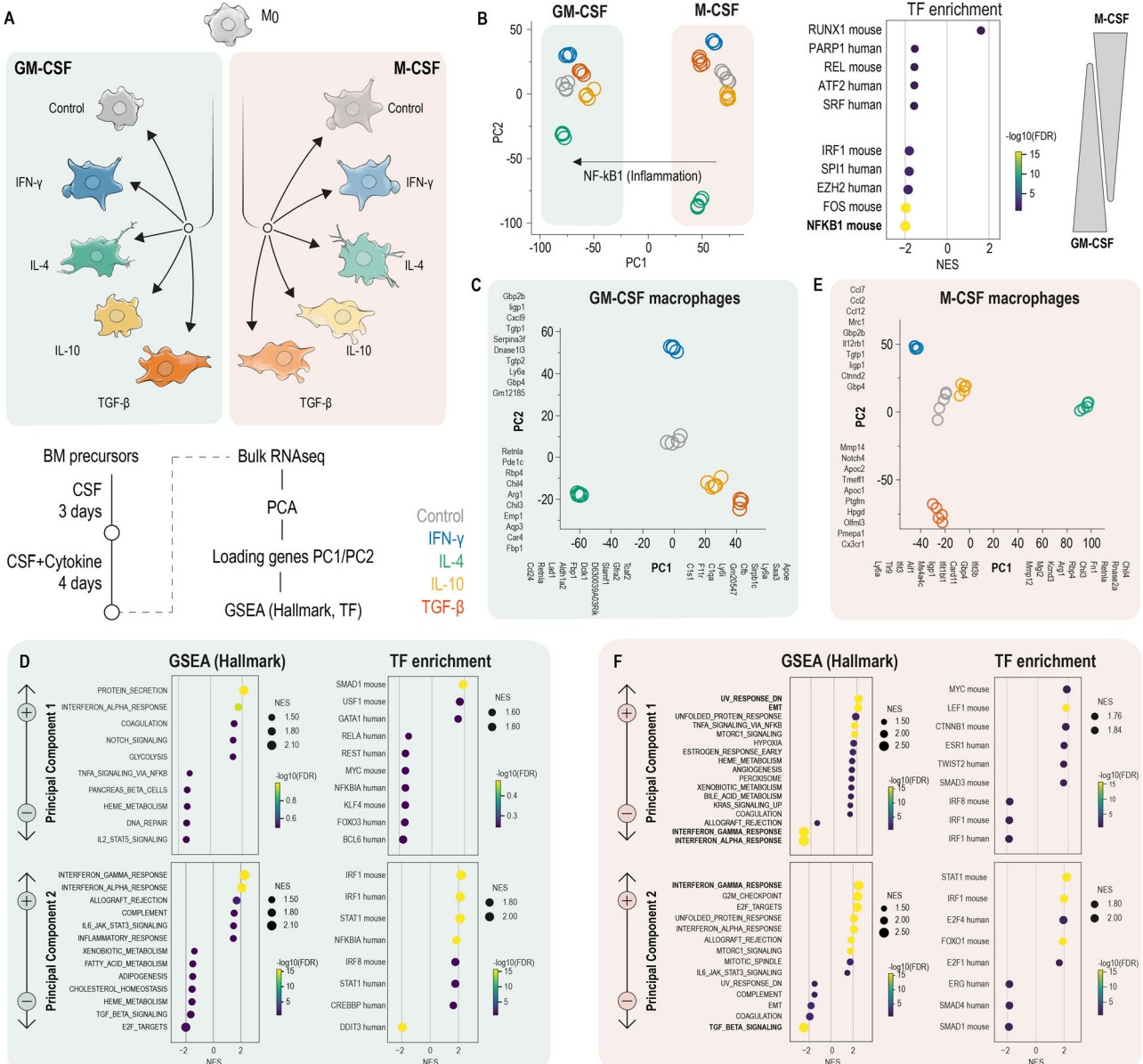

**Fig. 1 | Generation of reference macrophage states and transcriptomic hierarchy in M-CSF- or GM-CSF-derived macrophage lineages. A** Experimental workflow used to generate reference macrophage states. BM precursors were differentiated for three days with either GM-CSF or M-CSF and skewed with CSF + IFN-γ, IL-4, IL-10, or TGF-β, for additional four days yielding eight cytokine × ontogeny macrophage states. Sampling for bulk RNA-seq was performed on day 7. Color code: Control/unpolarized (gray), IFN-γ (blue), IL-4 (green), IL-10 (yellow), TGF-β (orange-red). **B** Joint principal component analysis (PCA) of GM-CSF- and M-CSF macrophages ($n$ = 5 biological replicates per condition, each dot represents one replicate). PC1 separates macrophages by ontogeny (GM-CSF < 0, M-CSF > 0); PC2 captures cytokine-induced variance. Transcription factor (TF) enrichment analysis of PC1 loading genes ($-\log 10$ FDR > 2) reveals predominance of NF-κB family transcription factors on the GM-CSF (PC1 < 0) side, confirming the intrinsic inflammatory bias of this lineage. **C** PCA of GM-CSF- macrophages ($n$ = 5 biological replicates per condition, each dot represents one replicate). PC1 (47%) separates IL-4-treated cells towards negative PC1; PC2 (25%) distinguishes IFN-γ-treated cells

(positive PC2) from IL-10/TGF-β states-treated cells (negative PC2). The top 10 genes contributing to positive loadings (right/top) and negative loadings (left/bottom) for each principal component are labeled on their respective axes. **D** GSEA using hallmark gene sets and transcription factor target genes for positive and negative loadings of PC1 (top row) and PC2 (bottom row) in GM-CSF macrophages. Dot size represents normalized enrichment score (NES); color intensity represents significance ($-\log 10$ FDR); only gene sets with FDR < 0.2 ($-\log 10$ FDR > 0.7) are shown. **E** PCA of M-CSF macrophages ($n$ = 5 biological replicates per condition, each dot represents one replicate). PC1 (54% of variance) separates IL-4-treated cells. The respective loading genes are associated with reparative programming; PC2 (26% of variance) spans an inflammatory (IFN-γ; positive) to remodeling (TGF-β; negative) continuum, with IL-10 occupying an intermediate position. The top ten genes contributing to positive and negative loadings for each PC are labeled on their respective axes. **F** GSEA using hallmark gene sets and transcription factor target genes for genes contributing to positive and negative loadings of PC1 (top row) and PC2 (bottom row) in M-CSF macrophages. Analysis parameters as in **D**.

(Fig. 4A). This system allows us to track how macrophage-cancer cell interactions evolve while maintaining controlled conditions. We selected M-CSF macrophages for this analysis because their homogeneous starting population enables cleaner resolution of tumor-induced transcriptional changes, and because our prior single-cell work with mixed spheroids

showed that tumor-cell transcriptional patterns closely mirror macrophage functional phenotypes (tumoricidal vs. non-tumoricidal)[7,22]. While parallel scRNA-seq of GM-CSF spheroids would provide additional resolution, our functional data (Figs. 5–7) directly demonstrate that ontogeny-dependent outcomes are robust and reproducible across both lineages. Importantly,

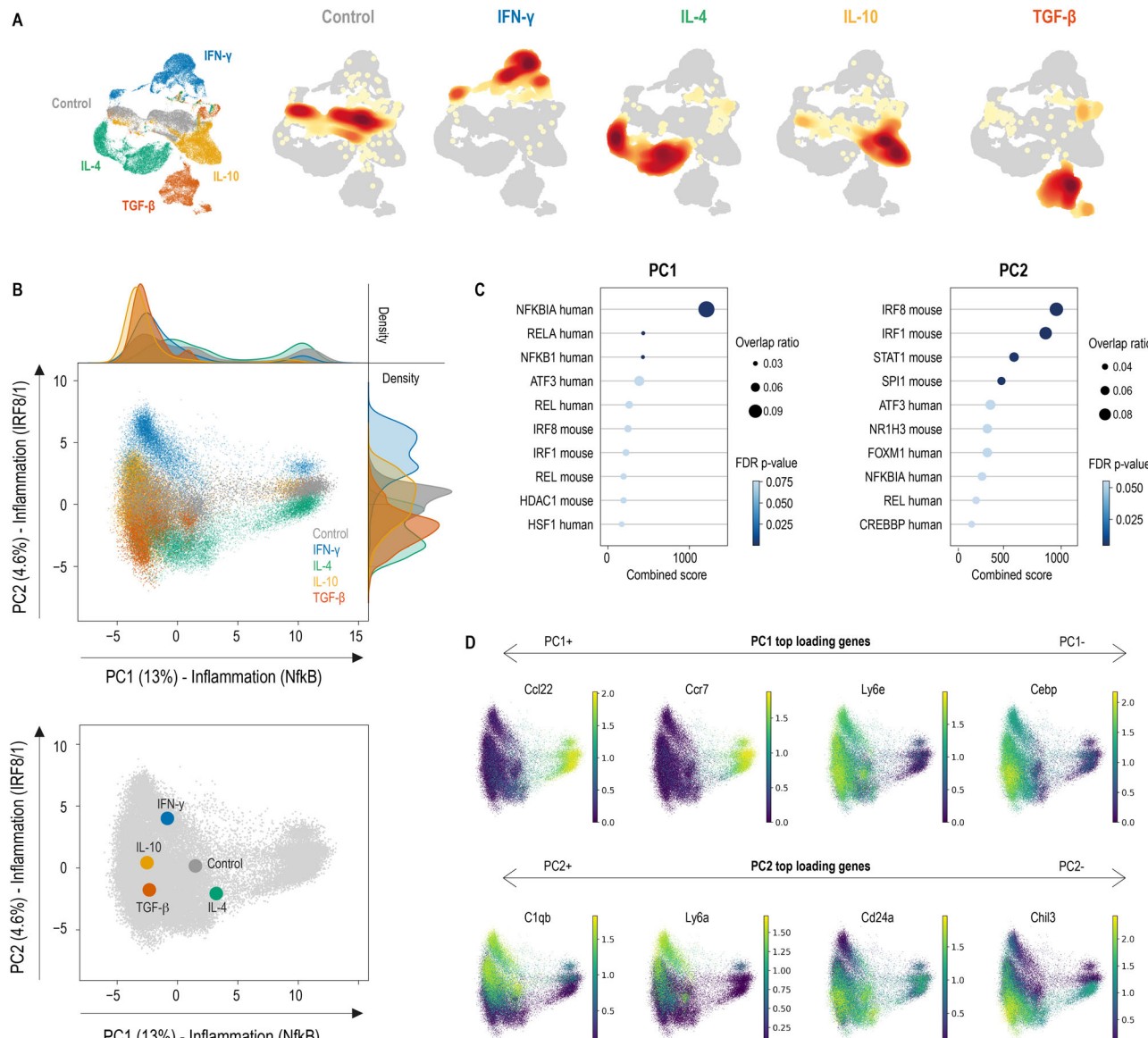

**Fig. 2 | Single-cell transcriptome analysis of GM-CSF macrophages.** BM precursors were differentiated with GM-CSF and polarized with IFN-γ, IL-4, IL-10 or TGF-β before multiplexed scRNA-seq (*n* = 49,441 cells: Control 10,051; IFN-γ 9,024; IL-4 10,834; IL-10 10,690; TGF-β 8,842). **A** UMAP of all GM-CSF macrophages, with each cell colored by cytokine treatment condition; Left panel: All cells displayed together showing overall distribution. Right panels: Individual UMAP plots split by treatment, with density projections (heatmap overlay) highlighting regions enriched for each cytokine-specific population. Treatment-specific spatial separation demonstrates distinct transcriptional states induced by each cytokine. **B** PC1 (13% of variance) captures an NF-kB-dominated inflammatory trajectory, with IL-4-treated cells separating in the positive direction; PC2 (4.6% of variance) captures an IRF-dominated interferon-response trajectory, with IFN-γ-treated cells clustering in the positive direction. Kernel-density estimates illustrate the distribution of cells along each PC, stratified by treatment. Treatment centroids are shown in principal component space. **C** Dot plots showing enrichment of TF target gene sets (from TRRUST database) among genes contributing to PC1 loadings (left panel) and PC2 loadings (right panel). Dot size represents the overlap ratio; dot color represents statistical significance (FDR *p*-value; darker colors indicate higher significance). Combined scores integrate overlap and significance. **D** UMAP plots of the two highest positive and negative loading genes for PC1 and PC2. Top row: The two highest positive (*Ccl22, Ccr7*) and negative (Ly6e, Cebpb) loading genes for PC1, demonstrating the NF-kB inflammatory axis. Bottom row: The two highest positive (C1qb, Ly6a) and negative (Cd24a, Chil3) loading genes for PC2, illustrating the interferon-response (positive) versus remodeling/trophic (negative) axis. Color scale indicates log-normalized expression.

macrophages were washed to remove cytokines before spheroid seeding; no polarizing cytokines were added thereafter, ensuring that observed effects resulted from stable macrophage programming rather than ongoing cytokine signaling. Confocal imaging confirmed that red-fluorescent macrophages from tdTomato⁺ mice integrate homogeneously within GFP-labeled tumor cells, forming compact and uniformly shaped spheroids (Fig. 4B).

We collected spheroids on days 1 and 5 and processed approximately 9000 spheroids per condition. After demultiplexing and quality control, 178,939 cells were retained for analysis (Fig. 4C). The temporal analysis revealed a compelling asymmetry in macrophage-cancer cell interactions. On day 1, macrophages occupied distinct transcriptional spaces, reflecting their prior cytokine imprints (Fig. 4D, left panel). By day 5, macrophages converged onto a single "tumor-educated" cluster, characterized by genes associated with anti-oxidative programs, resembling NRF2-stress TAMs[22], indicating that paracrine cues from tumor cells had overridden initial states (Fig. 4D, right panel, and Supplementary Fig. 3). Quantification by PCA confirmed this convergence: macrophage centroid dispersion decreased from 6.96 on day 1 to 3.18 on day 5 (mean pairwise Euclidean distance;

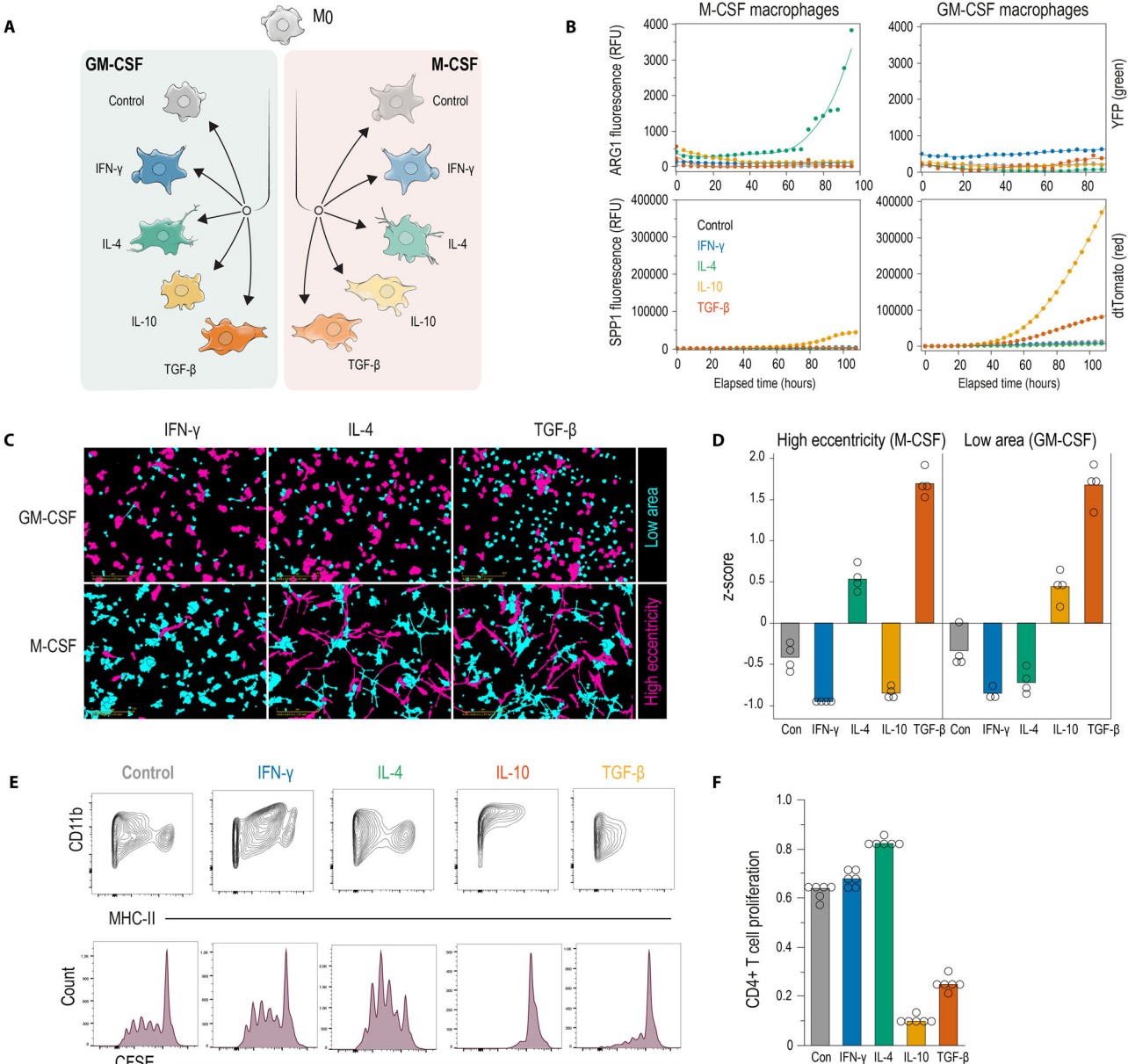

**Fig. 3 | Functional and phenotypic validation of the cytokine-ontogeny polarity switch. A** Experimental workflow used to generate reference macrophage. BM precursors were differentiated with GM-CSF or M-CSF and skewed with IFN-γ, IL-4, IL-10, or TGF-β, yielding eight cytokine × ontogeny macrophage states. Macrophages were harvested on day 7 for live-cell imaging, morphological analysis, and T-cell co-culture assays. Color code: Control (gray), IFN-γ (blue), IL-4 (green), IL-10 (yellow), TGF-β (orange-red). **B** Time-lapse live microscopy of Arg1-YFP (green) and Spp1-IRES-tdTomato (red) reporter macrophages. Curves plot mean fluorescence over 100 h for each cytokine in M-CSF and GM-CSF cultures (representative of three independent experiments; six imaging fields per well per condition were analyzed). Timepoint 0 represents the time of cytokine stimulation (day 3 of culture). IL-4 induces a progressive ARG1 signal only in M-CSF macrophages; IL-10 (and weakly TGF-β) induces SPP1 almost exclusively in GM-CSF macrophages; unpolarized and IFN-γ macrophages maintain low expression of both reporters throughout. **C** Representative fluorescence micrographs showing label-free cell classification based on automated quantification of cell area (primary metric for GM-CSF macrophages) and eccentricity (primary metric for M-CSF macrophages). Images were acquired using live-cell imaging (Incucyte) and analyzed using the

Advanced Label-Free Classification Module. Pseudo-color overlay indicates morphometric values: cyan = low (round cells with low area/eccentricity), magenta = high (elongated or large cells with high area/eccentricity). Representative images shown for each cytokine treatment in both M-CSF and GM-CSF cultures. Scale bar = 200 μm. **D** Z-scored classification data from four independent biological replicates (open circles). Eccentricity (primary morphological discriminant for M-CSF macrophages) and cell area (primary morphological discriminant for GM-CSF macrophages). Z-scores were calculated across all conditions to enable direct comparison. **E** Flow-cytometric contour plots for CD11b and MHC-II expression in GM-CSF macrophages across all treatment conditions and CFSE dilution histograms of OT-II specific CD4+ T cells co-cultured with peptide-pulsed GM-CSF macrophages. **F** Bar graphs of the cumulative data showing the fraction of OT-II CD4+ T cells that underwent proliferation. Data represent mean ± SEM from $n = 6$ independent cultures, one-way ANOVA Tukey–Kramer posttest corrected for multiple comparisons (Control vs IL-10 $p < 0.0001$; control vs TGF-β $p < 0.0001$; IFN-γ vs IL-10 $p < 0.0001$; IFN-γ vs TGF-β $p < 0.0001$; IL-4 vs IL-10 $p < 0.0001$; IL-4 vs TGF-β $p < 0.0001$. All other comparisons, including control vs IFN-γ, control vs IL-4, IFN-γ vs IL-4, and IL-10 vs TGF-β were not significant ($p > 0.05$; ns).

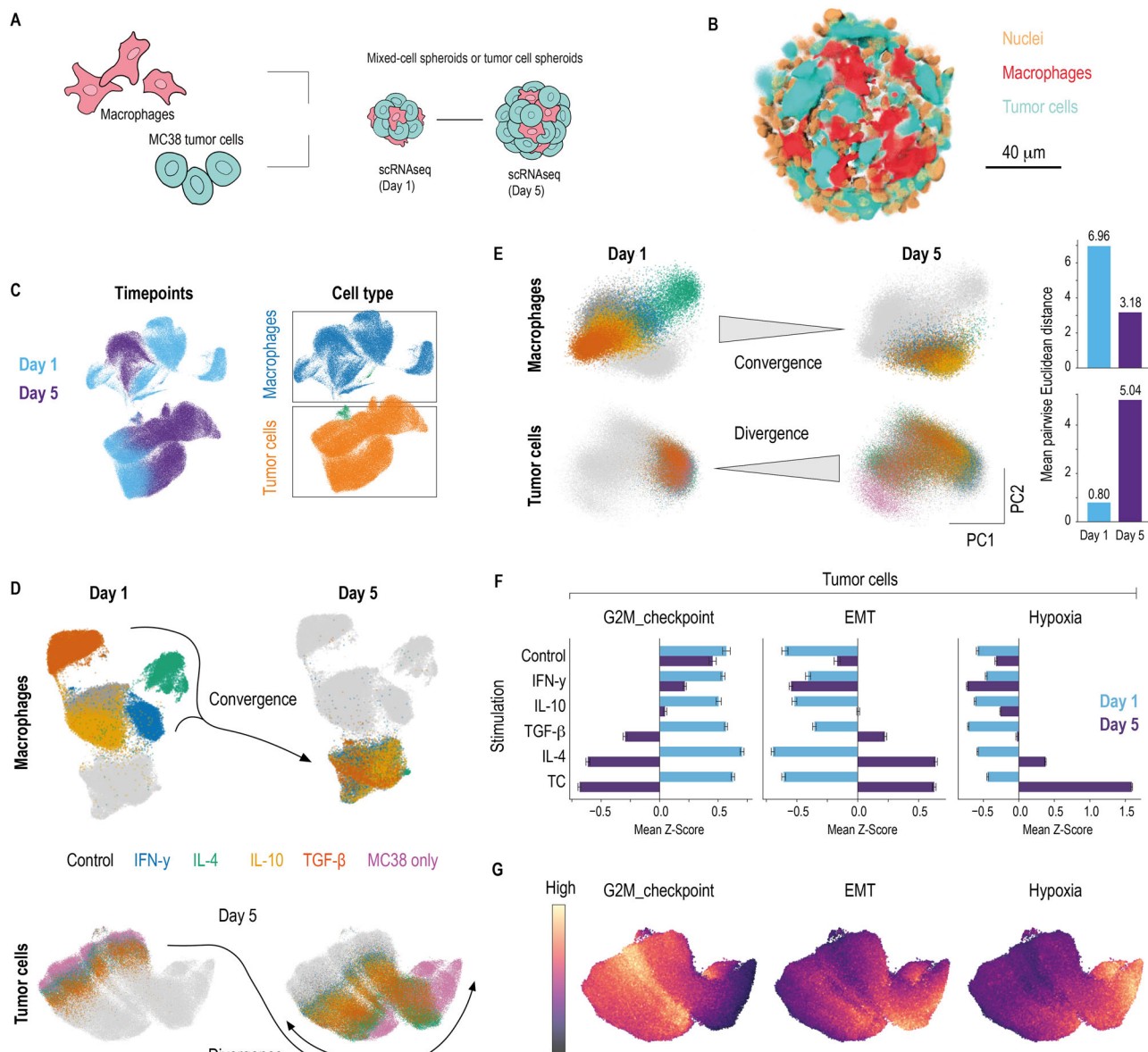

**Fig. 4 | Time-resolved single-cell mapping of mixed MC38–M-CSF macrophage spheroids. A** BM precursors were differentiated with M-CSF and skewed with IFN-γ, IL-4, IL-10, or TGF-β. On day 7, polarized macrophages were washed to remove cytokines and mixed 1:1 with GFP-labeled MC38 colorectal carcinoma cells in ultra-low-attachment microwell plates to form spheroids. Approximately 9000 spheroids per condition containing MC38 alone (tumor-only) or MC38 plus M-CSF macrophages (mixed) were harvested on days 1 (post-seeding) and 5 for pooled scRNA-seq. Sample sizes: total analyzed = 178,939 (Control day 1 13,145, control day 5 11,705; IFN-γ day 1 11,012, IFN-γ day 5 17,003; IL-4 day 1 11,700, IL-4 day 5 18,860; IL-10 day 1 14,978, IL-10 day 5 18,611; TGF-β day 1 17,026, TGF-β day 5 18,472, TC day 1 10,414, TC day 5 16,013). **B** 3D rendering from a z-stack of confocal micrographs acquired from a representative day-5 mixed spheroid. GFP-expressing MC38 tumor cells are shown in cyan; tdTomato+ IL-4-polarized M-CSF macrophages are shown in red; all cell nuclei (DAPI staining) are shown in yellow. The image demonstrates uniform integration of macrophages throughout the spheroid. Scale bar = 40 μm. **C** After demultiplexing and normalization, cells were plotted as UMAP, color-coded by sampling time and cell type: macrophages (blue) and tumor cells (orange). **D** UMAPs were recalculated for macrophages and tumor cells, showing cells from day 1 and day 5. Macrophage dynamics: On day 1, macrophages occupy cytokine-specific clusters reflecting their polarization states. By day 5, the clusters converge toward a "tumor-educated" transcriptional state. Tumor cell

dynamics: On day 1, tumor cells show no transcriptional variation regardless of macrophage co-culture condition. By day 5, tumor cell transcriptional states diverge in a macrophage-polarization-dependent manner. **E** To quantify transcriptional heterogeneity, we performed PCA separately for macrophages and tumor cells. For each condition, we computed the mean pairwise Euclidean distance in the PC1-PC2 space as a measure of transcriptional dispersion. Macrophages (top panels): Centroid dispersion decreases from 6.96 on day 1 to 3.18 on day 5, confirming transcriptional convergence. Tumor cells (bottom panels): Centroid dispersion increases from 0.80 on day 1 to 5.04 on day 5. Bar graph (right) shows quantification with numerical dispersion values. **F** Bar plots show mean *Z*-scores ±95% confidence intervals for gene set enrichment of selected modules (G2M Checkpoint, EMT, Hypoxia) in mixed-cell spheroid cells across different stimulation conditions (Control, IFN-γ, IL-10, TGF-β, IL-4) and MC38 spheroids (TCs). Gene set scores were calculated using AUCell for each individual cell, then aggregated by condition and timepoint. *Z*-scores were calculated within each gene set across all conditions and time points, with values representing pathway activity relative to the global mean (*Z*-score = 0). Bars are stratified by timepoint. **G** Feature UMAPs display per-cell gene-set scores for the same three modules (color intensity indicates module score). This spatial organization highlights the shift from uniform proliferation on day 1 to EMT in tumor cells co-cultured with anti-inflammatory macrophages by day 5.

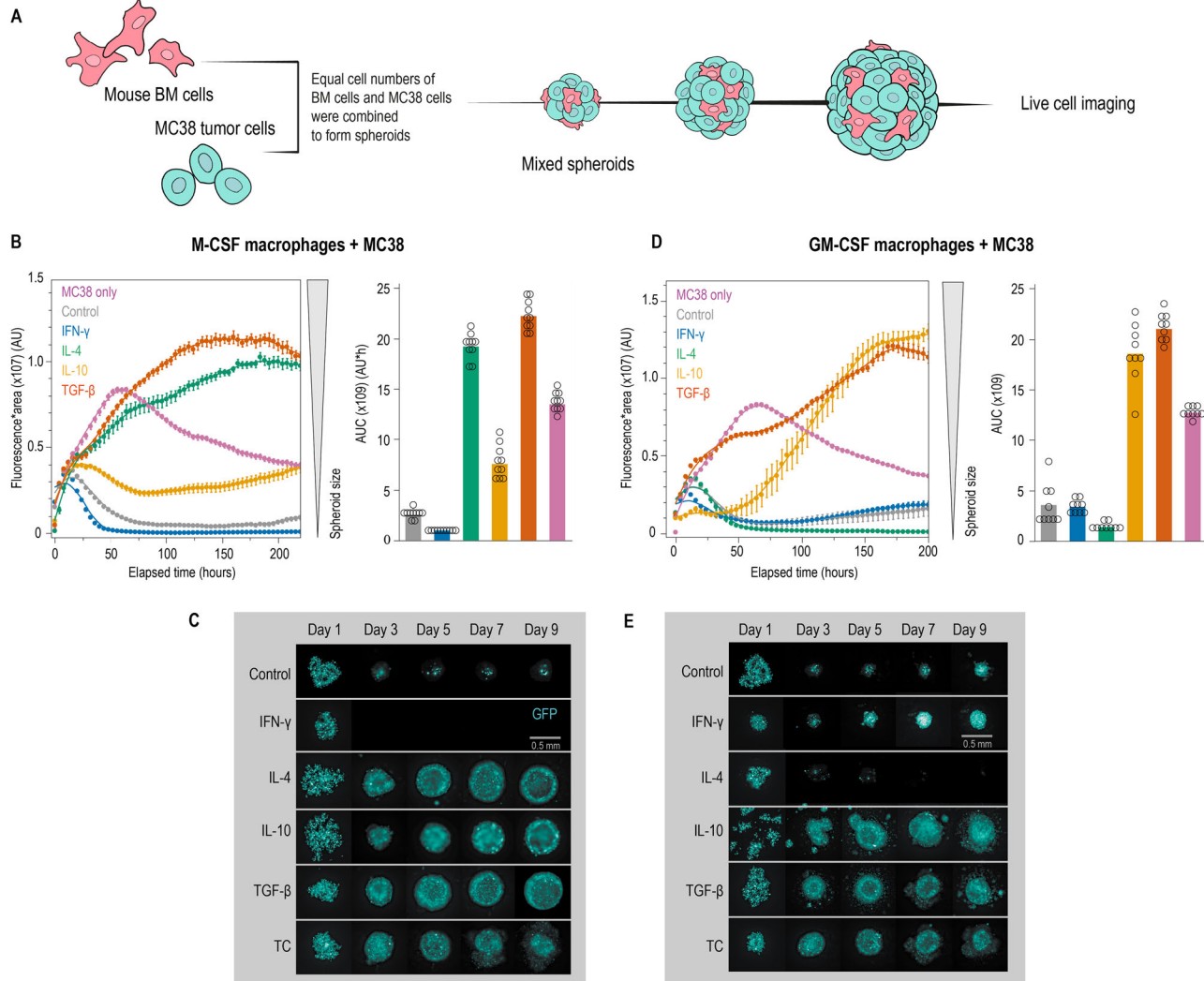

**Fig. 5 | Macrophage ontogeny and polarization dictate 3D tumor-spheroid growth. A** GFP-MC38 cells were cultured alone or mixed 1:1 with cytokine-polarized M-CSF or GM-CSF macrophages in ultra-low-attachment plates and imaged for 10 days. Macrophages were pre-polarized for 7 days as described in Fig. 1, then washed and mixed with tumor cells. Spheroids formed within 24 h and were imaged every 2–4 h for 10 days using an Incucyte S3 live-cell analysis system. GFP fluorescence intensity integrated across the entire spheroid area was used as a real-time readout of spheroid size. For each treatment condition 9–10 spheroids were analyzed. **B** M-CSF macrophages: Integrated GFP fluorescence intensities across the spheroid area over time (mean ± SE of 8–10 spheroids analyzed within one representative experiment) and area under the curve (AUC), reflecting cumulative spheroid growth over the 10-day period. Individual data points (open circles) show

replicate values. one-way ANOVA Dunnett posttest corrected for multiple comparisons (Control vs IFN-γ $p = 0.0110$, control vs all other cytokines $p < 0.0001$). **C** Representative fluorescence micrographs for each condition from the time course in **B**. Each image shows a single spheroid. Scale bar = 0.5 mm. **D** GM-CSF macrophages: Time-course analysis and AUC quantification identical to **B**. Data are the mean ± SE of 9 biologically independent spheroids from one independent experiment. Individual data points represent individual spheroids (Control vs IFN-γ $p = 0.9993$ (ns), control vs IL-4 $p = 0.0269$, control vs all other cytokines $p < 0.0001$). **E** GM-CSF macrophages: Representative fluorescence micrographs for each condition from the time-course experiment in **D**. Each row shows a representative spheroid over time. Scale bar = 0.5 mm.

Fig. 4E, showing PCA of macrophage populations with trajectory lines connecting day 1 to day 5 states[22].

In contrast, tumor cells showed the opposite pattern with increasing transcriptional dispersion from 0.80 on day 1 to 5.04 on day 5 (mean pairwise Euclidean distance of centroids) (Fig. 4E). To characterize these divergent tumor cell states, we scored three transcriptomic modules—proliferation (G2M_checkpoint), epithelial-to-mesenchymal transition (EMT), and hypoxia using GSEA (Supplementary Fig. 4)[7,22]. On day 1, nearly all tumor cells scored as proliferative regardless of macrophage polarization state. By day 5, the signatures diverged in a macrophage-polarization-dependent manner (Fig. 4F, G): Spheroids containing control or IFN-γ-polarized macrophages maintained high proliferative scores, whereas those co-cultured with IL-4- or TGF-β macrophages shifted toward high EMT scores. In contrast, tumor cells cultured without macrophages

acquired a strong hypoxia signature (Fig. 4F, G). IL-10 polarized macrophages induced intermediate states across all three modules. Notably, the apparent persistence of cell-cycle signatures in the spheroids containing IFN-γ-polarized macrophages indicates selective pressure rather than growth promotion: fewer tumor cells survive overall, but the remaining cells continue cycling, reconciling single-cell transcriptional programs with population-level outcomes[7,22].

Together, these data demonstrate that specific macrophage polarization states preferentially steer MC38 cells toward an EMT-like, low-proliferation, pro-invasive transcriptional state. Simultaneously, in the absence of continuous cytokine stimulation the tumor microenvironment erases macrophage polarization signatures over time, illustrating rapid and bidirectional cross-talk within the 3D spheroid niche. Thus, our findings reflect the combined effects of stable ex vivo macrophage programming and

emergent tumor-macrophage paracrine interactions that evolve over time in co-culture.

## The ontogeny-cytokine code determines tumor growth outcomes in 3D culture

To test whether our observed transcriptional and single-cell differences translate to functional consequences, we tracked spheroid growth for 10 days by live-cell microscopy with automated GFP fluorescence quantification (Fig. 5A). Spheroids composed of MC38 cells co-cultured with M-CSF macrophages containing control or IFN-γ-polarized cells rapidly lost GFP signal, reaching near-background levels within 3 days, whereby IFN-γ treatment showed the most potent tumor-killing activity (Fig. 5B, C). In contrast, spheroids with IL-4-or TGF-β-polarized macrophages showed a continuous increase in GFP fluorescence intensity, consistent with macrophage-mediated growth support. IL-10 produced an intermediate phenotype, with modest tumor growth comparable to control M-CSF macrophages. Tumor-cell-only spheroids initially gained fluorescence intensity but deteriorated after day 4, likely reflecting hypoxic stress in the absence of macrophage support (Fig. 5B, C).

Most remarkably, GM-CSF macrophage ontogeny completely reversed the IL-4 effect observed in M-CSF macrophages (Fig. 5D, E). In spheroids containing GM-CSF macrophages, IL-4 treatment now suppressed spheroid growth, while IFN-γ–polarized macrophages only partially suppressed growth, with outcomes comparable to control GM-CSF macrophages. In contrast, IL-10- or TGF-β-polarized macrophages promoted the most aggressive tumor growth. To establish the generalizability of this ontogeny-cytokine code, we tested two additional tumor cell lines. Similar outcomes were observed when we replaced MC38 cells with KP1.9 lung-cancer cells (Supplementary Fig. 5A, B). The reversal of IL-4 effects between M-CSF and GM-CSF macrophages was further validated in spheroids containing TC-1 lung carcinoma cells co-cultured with macrophages, providing evidence for a general ontogeny-dependent cytokine response polarity across different tumor cell types (Supplementary Fig. 5C, D).

Thus, our findings establish that CSF-dependent ontogeny determines whether cytokines promote or suppress tumor growth, with IL-4 showing the most fundamental reversal between lineages.

## The ontogeny-cytokine code governs cancer cell invasion in 3D matrix

To assess whether the ontogeny-cytokine code also controls tumor cell invasive behavior, we embedded spheroids in a laminin-rich matrix and quantified radial invasion over 72 h by automated image analysis (Fig. 6A). This assay revealed ontogeny-dependent invasion characteristics that paralleled the growth effects described above. In spheroids containing M-CSF macrophages (Fig. 6C), IL-4 and TGF-β treatments produced strong pro-invasive effects with rapid radial outgrowth far exceeding that observed with IFN-γ-polarized macrophages. IL-10 induced moderate invasive effects, mirroring its weaker growth-promoting activity. The invasive structures showed classic features of EMT-driven migration with finger-like projections extending into the matrix (Fig. 6B).

Consistent with the growth phenotypes, GM-CSF-derived macrophages reversed the cytokine rank order of invasion (Fig. 6D). IL-10 and TGF-β treatment elicited the strongest invasion, whereas IL-4—pro-invasive in M-CSF spheroids— lost this effect and produced minimal invasion, comparable to control macrophages. IFN-γ treatment again restrained invasion to levels below control macrophages. Quantitative analysis confirmed the phenotypic inversion between ontogenies (Fig. 6C, D).

These observations demonstrate that macrophage-derived signals regulate MC38 cell EMT and invasive capacity in an ontogeny-dependent manner that closely follows the pattern established for growth regulation.

## In vivo metastasis validates the ontogeny-cytokine code

To validate our in vitro findings in a physiologically relevant model, we performed intravenous injection of mixed-cell spheroids into C57BL/6 mice

and quantified lung metastatic burden after 3 weeks (Fig. 7A–C). Spheroids containing TGF-β–polarized macrophages produced the most substantial metastatic burden regardless of macrophage lineage, whereas spheroids with IFN-γ-polarized macrophages gave rise to sparse metastatic lesions. IL-4 effects depended on lineage: M-CSF macrophages seeded lungs densely, but GM-CSF spheroids failed to colonize following IL-4 polarization. IL-10 treatment phenocopied TGF-β with GM-CSF-derived macrophages, yielding strong metastatic burden, but was only weakly pro-metastatic in M-CSF.

These in vivo results are congruent with our in vitro findings, reinforcing the central principle that CSF ontogeny dictates whether IL-4 and IL-10 treatments act as pro- or anti-metastatic signals. Conversely, TGF-β and IFN-γ effects remained stable across both lineages, providing internal controls for the specificity of the ontogeny-dependent switching.

To synthesize our findings across multiple functional readouts, we integrated all functional parameters into a comprehensive ontogeny-cytokine code by z-score analysis. This integration revealed clear patterns across growth, invasion, and metastasis. IL-4 and IL-10 as the exemplars of ontogeny-dependent function have opposing z-scores in M-CSF and GM-CSF contexts (Fig. 7D, E). In contrast, TGF-β treatment consistently displayed protumor profiles, while IFN-γ retained a stable antitumor signature between ontogenies. Notably, the z-scores for T-cell activation activity in GM-CSF macrophages negatively correlated with the tumor progression markers across cytokine conditions, suggesting that enhanced T-cell stimulation contributes to antitumor effects in vivo. This quantitative framework—the ontogeny-cytokine code—offers a classification framework of macrophage function shaped by both developmental origin and environmental context.

## Discussion

This study establishes macrophage ontogeny as a master regulator that determines whether cytokine signals promote or suppress tumor progression. By applying a unified multiscale pipeline—tracking 3D spheroid growth, matrix invasion, and in vivo metastatic seeding—we directly compared how identical cytokines reshape macrophage function across ontogenies. The most striking finding is the CSF-dependent reversal of IL-4 function.

IL-4 exposure of M-CSF-primed macrophages induces an EMT-promoting, metastasis-enhancing program, aligning with evidence that IL-4Rα signaling in myeloid progenitors drives immunosuppressive myelopoiesis and tumor growth[23,24]. In contrast, IL-4 applied to GM-CSF-primed precursors yields an antigen-presenting, CCR7⁺ macrophage-DC spectrum that is intrinsically antitumor, even in the absence of further activation, extending prior studies[17,25,26]. We acknowledge that GM-CSF + IL-4 culture conditions are often used to generate monocyte-derived dendritic cells (moDCs), which retain macrophage markers while gaining antigen-presenting features, forming a macrophage/DC-like spectrum[17]. Our data demonstrate that IL-4 shifts GM-CSF macrophages along an NF-kB–dominated axis, which unexpectedly yields potent cytotoxicity—surpassing the semi-activated phenotype previously ascribed to GM-CSF + IL-4[25]. This ontogeny-dependent functional reversal aligns with recent evidence that GM-CSF can reprogram TAMs toward antitumor phenotypes[27,28].

IFN-γ drives STAT1/IRF programs promoting antitumor activity across both lineages. This lineage-independent response aligns with the strong TAM-activating function of IFN-γ[29] and with clinical observations that CXCL9-expressing TAMs and IFN-γ response signatures correlate with improved patient outcomes[19,30]. The consistent antitumor effects of IFN-γ across ontogenies suggest that its dominant STAT1 signaling may override ontogeny-specific chromatin states. The persistence of cell-cycle signatures in IFN-γ co-cultures likely reflects selection for cycling survivors rather than increased proliferation: fewer tumor cells remain overall, but the residual population is cycling[7,22].

IL-10 and TGF-β act as protumor drivers[31–34], most potent in GM-CSF–educated macrophages. Our findings provide functional evidence

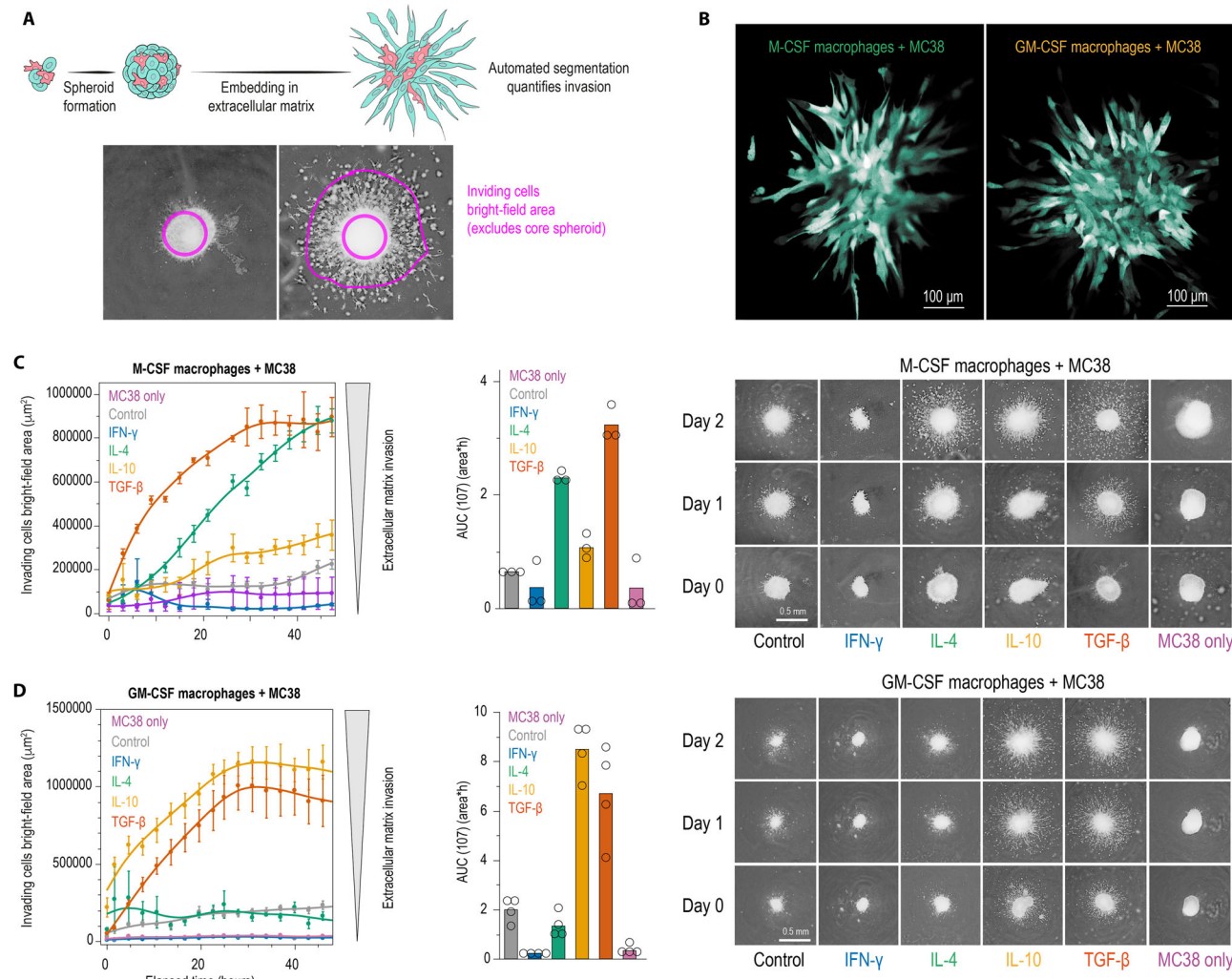

**Fig. 6 | Macrophage ontogeny and polarization dictate 3D tumor cell invasion.**
**A** GFP-MC38 cells were cultured alone or mixed 1:1 with cytokine-polarized M-CSF or GM-CSF macrophages in 96-well ultra-low-attachment or 24-well microwell plates. Three days post formation, spheroids were embedded in a laminin-rich matrix ($t = 0$ h) and invasion was visualized by Incucyte live-cell imaging or confocal microscopy. **B** Confocal micrographs acquired 24 h after embedding show dense, finger-like projections invading into the matrix in spheroids containing protumoral macrophages (IL-4 polarized M-CSF macrophages (green), IL-10 polarized GM-CSF macrophages (yellow)). Scale bar = 100 μm. **C** M-CSF macrophages: Quantitative spheroid invasion assay. Cell invasion was measured with an Incucyte live cell

imaging system every 4 h. The invading cell front was automatically segmented and quantified over time. Data are the mean ± SE of 3 biologically independent spheroids analyzed within one representative experiment. AUC quantification was analyzed by one-way ANOVA Dunnett posttest corrected for multiple comparisons (Control vs IFN-γ $p = 0.6900$ (ns), control vs IL-4 $p < 0.0001$, control vs IL-10 $p = 0.3239$ (ns), control vs TGF-β $p < 0.0001$, control vs TC $p = 0.6665$ (ns)). Representative inverted bright field images for each condition are shown. Scale bar = 0.5 mm. **D** GM-CSF macrophages: Analysis identical to **C** ($n = 4$ replicates per condition) (Control vs IFN-γ $p = 0.0714$ (ns), control vs IL-4 $p = 0.7959$ (ns), control vs IL-10 $p < 0.0001$, control vs TGF-β $p < 0.0001$, control vs TC $p = 0.0954$ (ns)). Scale bar = 0.5 mm.

that GM-CSF–educated macrophages can be re-wired by IL-10 into SPP1+, EMT-promoting, and metastasis-promoting states[31–34]. The IL-10-driven induction of SPP1 in GM-CSF macrophages is particularly notable, as SPP1+ TAMs have emerged as a key immunosuppressive population across multiple cancer types[21].

Our data argue against global TAM depletion[35] and instead support precision strategies that account for ontogeny-cytokine interactions[31,36]. Our integrated z-score analysis establishes a quantitative framework that categorizes macrophage functional responses into three classes: (1) Ontogeny-dependent switchers (IL-4): cytokine whose pro- versus antitumor effects reverse between M-CSF and GM-CSF contexts; (2) Ontogeny-independent tumor promoter (TGF-β): consistently protumor regardless of CSF; (3) Ontogeny-independent tumor suppressor (IFN-γ): consistently antitumor across both lineages. This classification may enable context-specific therapeutic predictions.

Several specific therapeutic approaches emerge: (1) In M-CSF-dominant tumors, IL-4 or IL-4Ra blockade could eliminate ARG1+

TAMs while preserving IFN-γ-responsive populations. (2) In GM-CSF-enriched tumors, targeting IL-10/TGF-β while delivering exogenous IL-4 could enhance antigen presentation and T-cell activation, exploiting IL-4's antitumor activity in the GM-CSF context. Moreover, engineered GM-CSF has demonstrated the capacity to reprogram TAMs toward antitumor phenotypes and potentiate responses to checkpoint inhibition and IL-12 immunotherapy in preclinical models, providing proof-of-concept for CSF-based macrophage reprogramming strategies[27].

Several limitations warrant consideration. Our reductionist approach excludes key tumor microenvironment components (vasculature, hypoxia, adaptive immunity) and tests only four cytokines, excluding prostaglandins, adenosine, heme, and other tumor metabolites known to modulate TAMs[37,38]. However, this simplified system was essential for establishing ontogeny-cytokine interactions without confounding variables. Finally, we limited time-resolved single-cell analysis in spheroids to M-CSF macrophages. Although GM-CSF macrophage dynamics were not profiled at single-cell resolution in 3D culture, the ontogeny-dependent effects on

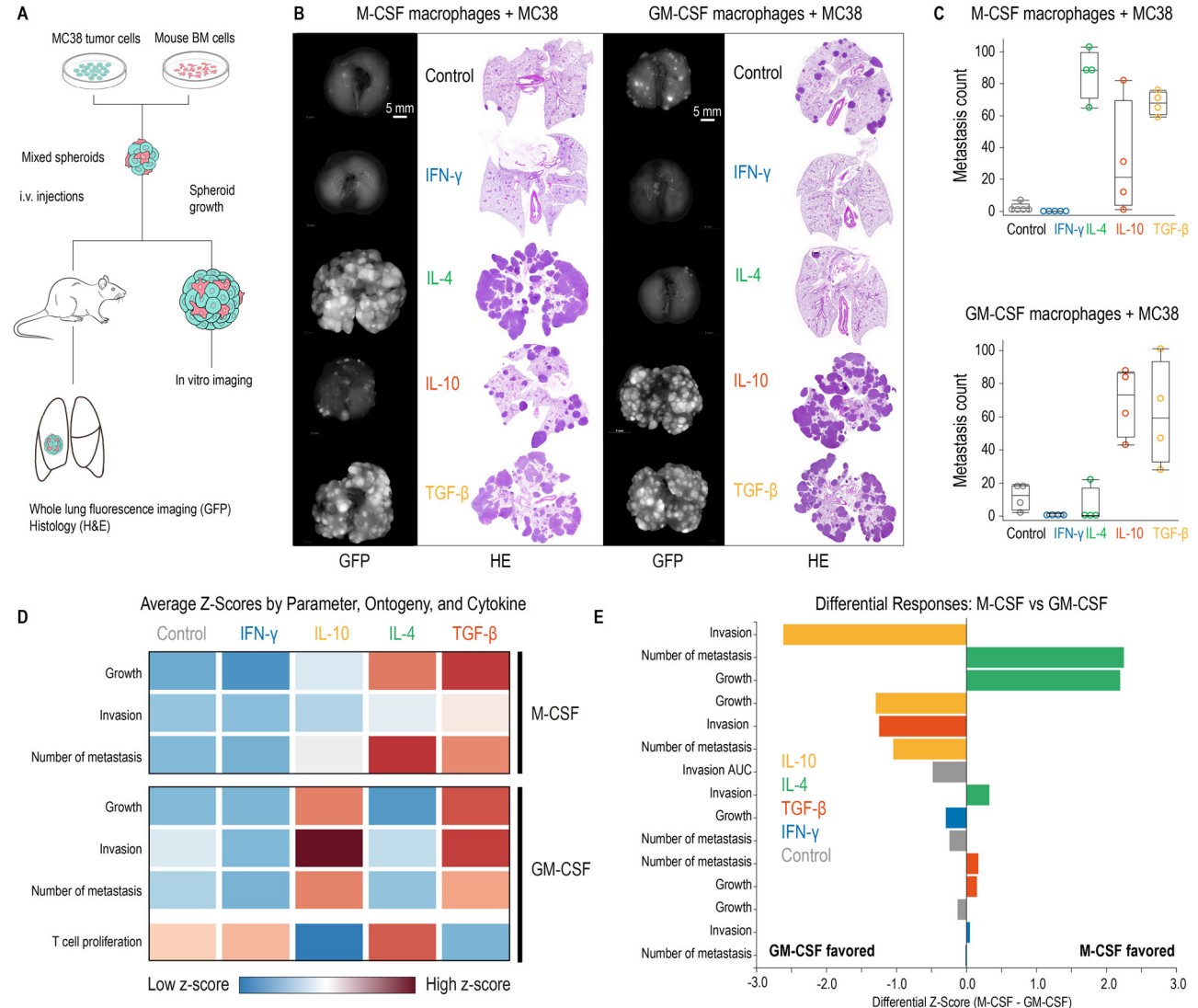

**Fig. 7 | Macrophage ontogeny and polarization dictate metastatic seeding in vivo.**
**A** Approximately 750 spheroids per condition were collected from microwell plates on day 4 post-formation and injected intravenously (i.v.) into C57BL/6 mice. Lungs were collected three weeks after injection (n = 4 mice per condition). **B** Metastases were visualized by histology (H&E) staining and whole lung fluorescence imaging. Scale bar = 5 mm. **C** Box plot showing metastatic nodule counts per representative lung section. Three blinded investigators independently counted metastases. Each dot represents the metastases count (number of nodules) of a representative lung section from one mouse (n = 4 mice per condition). ANOVA with Tukey–Kramer posttest corrected for multiple comparisons: **M-CSF macrophages:** (IL-10 vs IL-4 $p = 0.0020$; IL-10 vs TGF-β $p = 0.0492$; control vs IL-4 $p < 0.0001$; control vs TGF-β $p = 0.0002$; IFN-γ vs IL-4 $p < 0.0001$; IFN-γ vs TGF-β $p = 0.0001$. All other comparisons were not significant ($p > 0.05$; ns). **GM-CSF macrophages:** Control vs IL-10 $p = 0.0030$; control vs TGF-β $p = 0.0092$; IFN-γ vs IL-10 $p = 0.0006$; IFN-γ vs TGF-β $p = 0.0017$; IL-4 vs IL-10 $p = 0.0013$; IL-4 vs TGF-β $p = 0.0038$. All other

comparisons were not significant ($p > 0.05$; ns). **D** Integrated analysis of all functional read-outs. For each parameter, all values across treatment groups were converted to z-scores: $z = (x - \mu)/\sigma$, where $x$ is the individual measurement, $\mu$ is the mean across all conditions, and $\sigma$ is the standard deviation. This standardization enables direct comparison across functionally diverse assays with different scales and units, with positive z-scores indicating values above the parameter mean and negative z-scores indicating values below the mean. Heat-map shows average Z-scores for spheroid growth AUC, invasion AUC, lung metastasis count, and CD4-T cell proliferation for each cytokine within the M-CSF or GM-CSF lineage. **E** Differential Z-score plot (M-CSF minus GM-CSF z-scores for each functional parameter and cytokine) highlights cytokine effects that switch direction between ontogenies. Positive values (right) are M-CSF-favored, negative (left) GM-CSF-favored. IL-4 and IL-10 show the strongest polarity inversion, whereas TGF-β is consistently pro-tumor and IFN-γ consistently anti-tumor.

tumor-cell growth, invasion, and metastasis were reproducible across independent functional assays and consistent with the initial cytokine imprinting of each lineage.

Validation in more complex preclinical in vivo models and patient-derived systems remains critical for clinical translation. The temporal dynamics of CSF exposure in vivo likely differ from our ex vivo protocol, potentially affecting macrophage memory and plasticity. Our temporal analysis revealed that tumor-derived signals drive macrophage convergence toward an NRF2-dominated oxidative stress-response program, consistent with our previous identification of NRF2-stress TAMs as

an immunosuppressive, therapy-resistant phenotype[7,22,39]. However, the mechanisms by which ontogeny determines initial cytokine responsiveness remain incompletely understood. Future work combining epigenomic and metabolic profiling will be instrumental to define how CSF-dependent lineage priming shapes transcription factor accessibility and cytokine signaling outcomes[40–43].

Our data suggest that while tumor-derived environmental signals eventually drive macrophages toward a convergent transcriptional state in spheroids, the initial cytokine-imprinted functionality determines the trajectory of tumor growth or suppression. This implies that the ontogeny-

cytokine code acts as a critical gatekeeper during the early phases of tumor-macrophage interaction, setting a functional course that persists even as transcriptomic profiles begin to align under environmental pressure.

Beyond cancer, our findings have implications for understanding macrophage function in other diseases. The ontogeny-cytokine code may apply to inflammatory diseases, wound healing, and metabolic disorders where macrophage polarization plays crucial roles, as developmental origin determines cytokine response polarity across diverse physiological and pathological contexts[44–47].

The ontogeny-cytokine code presented herein offers a framework for understanding TAM functional heterogeneity and designing precision immunotherapy. Assessing the relative dominance of M-CSF or GM-CSF signaling within tumors could inform personalized cytokine-modulating strategies. Integrating CSF and cytokine signatures into patient stratification frameworks could thus refine macrophage-targeted interventions in cancer therapy.

## Methods
Providers, catalog and product numbers are listed in the Supplementary Information file.

### Mice
Male and female mice aged 8–12 weeks were used in this study. C57BL/6 mice were obtained from Charles River Laboratories. To generate *tdTomato*[+] macrophages, *Vav-Cre* mice (Swiss Immunological Mouse repository, SwImMR) were bred with *Ai14*[Rosa26-LSL-tdTomato] mice (The Jackson Laboratory). B6.Cg-Tg(TcraTcrb)425Cbn/J (OT-II) were obtained from the SwImMR. Spp1-IRES-tdTomato and Arg1-ires-YFP mice were obtained from the Jackson Laboratories. For lung tumor studies, mice were randomly allocated to treatment groups, and the investigators were blinded to allocation during experiments and outcome assessment.

### Cell lines
GFP-MC38 (donated by Gerhard Christofori, Department of Biomedicine, University of Basel, Basel, Switzerland) was cultured in RPMI-1640 medium supplemented with 10% fetal bovine serum (FBS, Gibco), 1% penicillin/streptomycin (P/S, Gibco), 1% non-essential amino acids (NEAA, Gibco), and 1% sodium pyruvate (Gibco). Tumor cell line culture KP1.9 (donated by Mikael Pittet, Ludwig Cancer Research, University of Geneva, Switzerland) was cultured in IMDM supplemented with 10% FBS, 1% P/S, and transduced with a GFP-expressing lentiviral vector (Sartorius) before use in spheroid assays. TC-1 cells (Cytion) were cultured in IMDM supplemented with 10% FBS, 1% P/S and transduced with a mScarlet fluorescent protein lentiviral vector (VectorBuilder).

Cell lines with homogeneous fluorescence expression were obtained by FACS sorting. Cell line authentication was performed before and after cell sorting by Short Tandem Repeat (STR) DNA genotype analysis (Microsynth, Balgach, Switzerland). Cell cultures were maintained at 37 °C and 5% $CO_2$ in a humidified incubator. All cells used in this study were confirmed to be negative for mycoplasma.

### Primary BM cultures
BM cells were isolated by flushing the femurs and tibias of 8- to 12-week-old C57BL/6 mice and then passed through a 70-μm filter. The BM cells were plated at a density of $3 \times 10^5$ cells/ml on tissue culture-treated 60 mm UpCell dishes (Nunc™ UpCell™, Thermo Fisher Scientific) in complete RPMI-1640 medium (10% fetal calf serum (FBS), 1% L-glutamine, and 1% P/S) supplemented with 100 ng/ml recombinant mouse M-CSF (PeproTech) or 20 ng/ml recombinant GM-CSF (PeproTech) as previously described[18]. On day 3, half of the medium was replaced. Cultures were treated on day 3 with IL-4 (20 ng/ml, PeproTech), IL-10 (100 ng/ml), TGF-β (100 ng/ml), or IFN-γ (20 ng/ml, PeproTech). The BM cells were harvested for analysis on day 7 from the temperature-responsive cell culture plates after cooling to room temperature. Cells were washed twice in phosphate-buffered saline (PBS) and centrifuged ($300 \times g$, 10 min) before spheroid formation.

### Label-free cell classification analysis
At the end of the differentiation period, BM cells were gently detached and reseeded at 20,000 cells per well in 96-well clear-bottom plates (TPP). Bright-field images were acquired on an Incucyte® S3 (20× objective). Segmentation and single-cell morphometry (eccentricity, projected area) were performed using the Incucyte® Advanced Label-Free Classification Module (Sartorius). At least 6 fields per well and 4 replicates per condition were quantified.

### CD4+ T cell isolation and CFSE labeling
Lymphocyte T cells were positively enriched from spleen single-cell suspensions using CD4 enrichment kit (Thermo Fisher Scientific) according to the manufacturer's instructions. Isolated CD4+ T cells were labeled with CFSE (Thermo Fisher Scientific) at RT for 20 min, washed with RPMI Medium and counted before use. The final purity, confirmed by flow cytometry, was >95%.

### OT-II assay
A total of $2 \times 10^4$ GM-CSF- stimulated BM cells were plated in 96-well round-bottom plates (Falcon), pulsed or not with 1 μg/ml OVA[323–339] peptides (Sigma) for 45 min at 37 °C, and washed three times with PBS. Subsequently, $1 \times 10^5$ CFSE-labeled naive CD4+ T cells isolated from spleens of OT-II mice were added to the BM cells in complete RPMI-1640 medium and co-cultured at 37 °C. CFSE dilution was assessed by flow cytometry after 3 days.

### 3D tumor spheroid production, culture
Single-spheroid culture: $5 \times 10^3$ GFP-MC38 or KP1.9 cells ± macrophages were seeded in 100 μl tumor cell culture medium in 96-well Ultralow Attachment Plate PrimeSurface® 3D Culture Spheroid plates (S-BIO). Multispheroid culture in microwell plates: GFP-MC38 cells ($5 \times 10^4$) ± macrophages (at a 1:1 ratio) were seeded in 800 μl of tumor cell medium with M-CSF (100 ng/ml) in a 24-well SphericalPlate® 5D microwell (Axonlab). 800 μl of fresh culture cell medium was added on day 3.

### 3D tumor spheroid analysis
Single spheroids were imaged with an IncuCyte S3 instrument (Sartorius). The area and fluorescence intensities of the images were measured using the IncuCyte Spheroid Software Module (Sartorius). Data are reported as spheroid fluorescence intensity integrated across the spheroid area (for tumor cells expressing a fluorescent protein) or as spheroid area. For the spheroid invasion assay, a mask based on the invading cell area was created automatically with the IncuCyte Spheroid Software Module. Each experiment included $n = 8–10$ biologically independent spheroids per condition. Experiments were repeated independently at least twice with comparable results; representative experiments are shown.

### High-resolution imaging of spheroids for visualization of EMT-like invasion
On day 3 post spheroid formation, spheroids were transferred from the microwell plate (Axonlab) to a flat glass-bottom plate (TPP) and embedded into Cultrex extracellular matrix (Bio-Techne). After 24 h, spheroid morphology and matrix invasion of cancer cells were imaged using the laser-scanning microscope (CLSM Leica Sp8-inverse). Confocal micrographs with cell type-specific fluorescence were acquired as z-stacks using a PL APO CS 20x objective. Images are presented as Z-projection (sum of slices) using ImageJ/Fiji.

### High-resolution imaging of spheroids for visualization of cellular composition
On day 5 post spheroid formation, spheroids were collected from the microwell plate (Axonlab) in Eppendorf tubes, washed once for 2 min with 1X PBS, subsequently nuclei were stained using NucBlue™ for 1 h. Spheroids were then washed three times with 1X PBS and mounted on SuperFrost plus slides with ProLong™ Gold Antifade mountant (Thermo

Fisher Scientific, P36930). A coverglass was placed on top, and specimens were imaged immediately. Spheroid morphology was assessed using the laser-scanning microscope (CLSM Leica Sp8-inverse). Confocal micrographs with cell type- and nucleus-specific fluorescence were acquired as z-stacks using a HC Pl APO CS2 63x/1.4 oil objective. 3D stacks were rendered in Bitplane Imaris (Oxford Instruments).

## Spheroid digestion

Spheroids were dissociated in 2 ml digestion medium (RPMI medium (Gibco) + 25 µg/ml Liberase™ (Roche) + 40 µg/ml DNase I (Roche; 2000 µ/ml) and incubated for 30–45 min in a water bath at 37 °C with gentle shaking every 5 min. Then, 4 ml PBS + 0.04% BSA was added to stop the digestion. Digested spheroids were used immediately.

## Lung metastasis model in mice

Approximately 750 spheroids were collected from microwell plates (equal to the content of one macrowell) and injected intravenously into the tail of C57BL/6 mice. Three weeks post-injection, the lungs of anesthetized mice were perfused with PBS through the right ventricle and the trachea and collected for whole organ fluorescence imaging with a Zeiss Discovery V8 stereomicroscope and histology.

## Bulk RNA sequencing workflow and analysis

RNA was extracted from macrophages using the RNeasy Mini kit (Qiagen) according to the manufacturer's protocol, including on-column DNase I treatment. RNA quality was validated on an Agilent Technologies 4150 Tapestation using RNA Screentapes, and only samples with an RNA integrity number (RIN) of >9 were used for sequencing. cDNA libraries were generated at the Functional Genomics Center Zurich (FGCZ) from RNA samples using the Illumina Stranded mRNA Prep ligation kit following the manufacturer's instructions. The quality and concentration of the libraries were determined using an Agilent Technologies 4200 Tapestation with DNA Screentapes. The libraries were pooled in equimolar amounts and sequenced on an Illumina NovaSeq X Plus sequencer (paired-end 150 bp) with a depth of at least 20 million reads per sample. Reads were aligned to the reference genome Mus_musculus/GENCODE/GRCm39/Annotation/Release_M37-2025-07-03. The quality of alignment was evaluated using Samtools/1.20. Counts were obtained using the featureCounts function of the Rsubread package (v1.22.2)[48]. Differential expression analysis was performed with the DESeq2 R package (v1.26.0)[49].

## scRNA-seq sample preparation and analysis

For libraries generated with the 10X Genomics Next-GEM Single Cell Fixed Kit the following workflow was used. Single-cell suspensions were fixed following the demonstrated protocol Fixation of cells & Nuclei for Chromium Fixed RNA Profiling (10X Genomics, CG000478) and processed for long-term storage at −80 °C. After storage, up to 2 Mio cells per sample were hybridized with unique single Mouse WTA probes using BC001-004 according to the User Guide Chromium Fixed RNA Profiling Reagent Kits for Multiplexed Samples (10X Genomics, CG000527). For libraries generated with the newer 10X Genomics GEM-X Flex Kit the following workflow was used. Single-cell suspensions were fixed following the demonstrated protocol Fixation of cells & Nuclei for GEM-X Flex Gene Expression (10X Genomics, CG000782) and processed for long-term storage at −80 °C. After storage, a maximum of 0.5 Mio cells per sample were hybridized with unique single Mouse WTA probes using BC001-004 according to the User Guide GEM-X Flex Gene Expression Reagent Kits for Multiplexed Samples (10X Genomics, CG000787). Following gene expression library construction, ready-made libraries were sequenced at the FGCZ on an Illumina NovaSeq X Plus system following the recommendations of 10X Genomics. Downstream analysis was performed in Python (version 3.12.10) with Scanpy (1.11.1)[50].

## Read alignment

Reads were aligned to the mouse reference genome Ensembl GRCm39 (Release_M31-2023-01-30) using 10x Genomics Cell Ranger v9.0.0.

## Quality control and preprocessing

To assess the quality of the cells, the following covariates were considered: number of genes expressed in a cell (n_genes_by_counts), number of counts per cell (total_counts), and percentage of mitochondrial RNA (pct_counts_mt). Cells that expressed fewer than min_genes or more than max_genes were filtered out. Cells with a percentage of mitochondrial RNA greater than max_pct_mt were considered dead and removed from the analysis. Genes that were expressed by fewer than min_cells cells were excluded. See below for the cutoff values used in each experiment. The count data were normalized by an algorithm based on deconvolving size factors from cell pools implemented in the R package scran (calculateSumFactors)[51] and $\log(x+1)$ (sc.pp.log1p) transformed, yielding normalized expression values.

## Data integration

Multiplexed samples were merged into one dataset by simple concatenation. Additionally, samples from different experiments were integrated using the harmony algorithm (sc.external.pp.harmony_integrate)[52] after normalization and PCA.

## Dimensionality reduction and clustering

For dimension reduction, the following steps were performed using the Python package Scanpy: identifying highly variable genes (sc.pp.highly_variable_genes), performing PCA using highly variable genes (sc.tl.pca), computing the neighborhood graph (sc.pp.neighbors) and computing the UMAP (sc.tl.umap).

## Cell type annotation and functional classification

To identify cell types, we analyzed the expression of marker genes and other differentially expressed genes (sc.tl.rank_genes_groups with method = 'wilcoxon'). Depending on the experiment, we analyzed these genes separately or by scoring gene sets using the algorithm *AUCell*[53] implemented in the package decoupleR[54] (version 1.8; dc.run_aucells). GSEA was performed to assess functional and biological process-related differences between clusters or conditions. First, genes were ranked using the output of the Wilcoxon rank-sum test (rank = −log10(adj. *p* value)*sign(logfoldchange)) and then fed to the GSEA algorithm implemented in the Python package decoupleR[54] (dc.get_gsea_df), resulting in a normalized enrichment score (NES) and a false discovery rate (FDR) per gene set. For transcription factor analysis, gseapy.enrichr with the TRRUST_Transcription_Factors_2019 gene set database was used.

## Flow cytometry

Cells were preincubated with Mouse BD Fc Block™ (≤1 µg/million cells in 100 µl, BD Biosciences) at 4 °C for 10 min. Anti-I-A/I-E (clone M5/114.15.2) was purchased from BD Biosciences. Anti-CD4 (clone GK1.5) and anti-CD11b (clone M1/70) antibodies were purchased from BioLegend. Corresponding isotype-matched irrelevant specificity controls were purchased from BD and BioLegend. Multiparameter analysis was performed with an LSRFortessa analyzer (BD Biosciences). The data were analyzed using FlowJo software (version 10.7.1).

## Histology

Mice were anesthetized by intraperitoneal injection of ketamine (80 mg/kg), xylazine (16 mg/kg), and acepromazine (3 mg/kg) and transcardially perfused with cold PBS. Lungs were fixed in 10% neutral-buffered formalin for 24 h, transferred to 70% ethanol, embedded in paraffin, and sectioned at 10 µm for hematoxylin and eosin (H&E) staining.

## Microscopy image acquisition and analysis

Whole-lung-tumor sections were imaged using a Zeiss Axio Scan Z1 slide-scanner microscope. Whole spheroids were imaged using a Leica SP8 confocal laser-scanning microscope. Images were analyzed using QuPath and ImageJ. For single-channel fluorescence images, brightness, contrast, and color balance were adjusted using Adobe Lightroom (version 8.4) with

**Article**

identical settings applied to all images within each experiment. No nonlinear image processing was performed.

## Statistics and reproducibility

Statistical analyses were performed using Prism 10 (GraphPad) and JMP 18 (SAS). Unless otherwise stated, comparisons among multiple groups were analyzed using one-way ANOVA followed by Tukey–Kramer or Dunnett's post hoc test to correct for multiple comparisons, as indicated in the figure legends. Exact $p$ values are reported in the figure legends.

Biological replicates were defined as independent cell cultures or individual animals. The number of biological replicates ($n$) for each experiment is indicated in the corresponding figure legends. In vitro experiments were independently repeated at least two times with similar results unless otherwise stated. In vivo experiments were performed with the indicated number of animals per group. Data are displayed as individual data points with summary statistics to visualize the distribution of the data. $Z$-scores were calculated separately for each parameter (Growth AUC, Invasion AUC, Number of metastases, and T cell proliferation) to standardize values across experimental conditions. For each parameter, values across treatment groups were pooled to calculate the population mean ($\mu$) and standard deviation ($\sigma$). Individual $z$-scores were computed as $z = (x - \mu)/\sigma$, where $x$ represents an individual measurement. Positive $z$-scores indicate values above the parameter mean and negative $z$-scores indicate values below the mean.

## Study approval

We have complied with all relevant ethical regulations for animal use and all animal experiments were performed according to animal experimentation licenses as approved by the Swiss Federal Veterinary Office.

## Reporting summary

Further information on research design is available in the Nature Portfolio Reporting Summary linked to this article.

## Data availability

Bulk RNA-seq and scRNA-seq data have been deposited in the Gene Expression Omnibus (GEO) under accession number GSE304868[55]. Source data underlying all graphs and charts in the main figures are provided as Supplementary Data 1. All other data supporting the findings of this study are available from the corresponding author upon reasonable request.

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

## Acknowledgements
We thank G. Christofori for providing GFP-MC38 and M. Pittet for providing KP1.9 cells. Transcriptome and scRNA sequencing was performed at the Functional Genomics Center Zurich (FGCZ) of the University of Zurich and ETH Zurich. This work has been funded by the Swiss National Science Foundation (project grant 310030_201202/1 and 320030-232113) to FV, Swiss Cancer Research foundation (project grant KFS-5944-08-2023) to FV, Carigest SA to FV, Vontobel, Novartis, Maiores, and Wilhelm Sander foundations to FV.

## Author contributions
Conceptualization, F.V.; methodology, D.J.S., F.V.; investigation, N.S.L., L.B., M.E., F.V.; formal analysis, D.J.S., M.J.P., F.V.; visualization, D.J.S., R.H., F.V.; writing, D.J.S., F.V.; funding acquisition, F.V.

## Competing interests
The authors declare no competing interests.

## Declaration of generative AI and AI-assisted technologies in the writing process
AI language models (ChatGPT-5, OpenAI; Claude Opus 4.5) were used for language editing; all scientific content was written and verified by the authors. No generative AI images were used.
