## [Transparent Peer Review file · Communications Biology]

An ontogeny-cytokine code determines macrophage response polarity and tumor outcomes

Corresponding Author: Professor Florence Vallelian

Version 0:

Reviewer comments:

Reviewer #1

(Remarks to the Author)

This is an interesting study by Schaer et al., reporting data supporting that the tumor inhibitory vs supportive roles of macrophages are a function of both the developmental origin and the cytokines in the microenvironment of these cells. For example, here authors show that IL-4 induces tumor promoting functions in macrophages grown with M-CSF while it induces tumor inhibitory functions in macrophages grown with GM-CSF. Using bulk and single cell RNA-seq, authors investigated the transcriptome of the 8 macrophage states they generated through different combination of M-CSF or GM-CSF with various cytokines. Authors further validated the functions of each macrophage state through both in vitro co-culture (3D spheroid culture) and in vivo experiments. Overall this is a well written manuscript with valuable data. Yet, some of the data reported here still seem preliminary and sections are underdeveloped, including the single cell data analysis. Extended validation of the functions of the macrophage states should be performed using other tumor cell lines. Some experiments were performed with M-CSF macrophages, others with GM-CSF macrophages: authors should keep experiments consistent throughout. In addition, further work on the mechanisms underlying the response of macrophages to the different CSF/cytokine combinations would be needed. Authors should also clarify the potential clinical application / impact of their data. See details below:

- Overall the results section should be expanded with further and more precise description of the figures. Reference to figures is also not always clear.
- Figure 1: 'IL-10 shifted modestly toward the IFN-low quadrant' = this is not clear in Figure 1E. Also lfi44 doesn't seem to be in any of the panels of Figure 1. GSEA and TF data are hardly discussed.
- Figure 2: IL-4 GM-CSF macrophages are in fact split in 2 clusters, with only one of these 2 clusters enriched for NF-kB controlled gene expression. 'TGF-B and IL-10 co-clustered...inflammatory pole' (p.6): this seems to be true for TGF-B but not for IL-10 cells. Violin plots showing expression of specific genes in each cell population would help here. Was single cell RNA-seq also performed on M-CSF macrophages? In addition, observations made from single cell RNA-seq data should be validated (expression of specific gene marker for instance).
- Figure 3: 'consistent with previous observations' (p.6) = can you please clarify here? Figure 3E and F= these figures are hard to read, quantification, labels and stats are missing. Why weren't these experiments performed on M-CSF cells?
- Figure 4-5: what's the rationale for performing these experiments at day 1 and 5 of coculture?
- Figure 4: Were similar experiments performed using GM-CSF macrophages? + 'Importantly,... rather than ongoing cytokine signaling' = authors should consider what tumor cells and macrophages will produce and secrete in culture. Figure 4E: it's not clear from these data what happens to the macrophages in culture, how different are they from day 1, any interesting differently genes to show here, or gene enrichment analysis between day 1 and day 5? From what data show here it seems that all macrophages, regardless of their original phenotype, adopt the same phenotype when in culture with cancer cells. It's not clear how the authors reach the following conclusion 'Notably, the apparent persistence... outcomes' re IFN- γ conditions. Overall this experiment would need to be repeated with GM-CSF macrophages and different tumor cell lines. Authors should also investigate specific markers to validate their single cell RNA-seq data.
- Figure 5 = authors should validate their comments on hypoxic stress. KP1.9 cancer cells should also be used in co-culture with M-CSF cells.
- Figure 7D – E: It's not clear what the point of these panels is. Authors should clarify the potential clinical impact of these data. In discussion authors state 'Our data argue against global TAM depletion' which is true but it doesn't come across clearly enough in the manuscript, including in the results section. Further experiments aimed at supporting this statement could be performed. Authors should further clarify how their data could help patient treatment and care, and for which cancers specifically.

- Figure 3B – 5B-D – 6C-D = it's not clear what these graphs show. 'representative of three independent experiments' = can you please clarify what this means? Were any statistical tests run on these experiments?

Reviewer #2

(Remarks to the Author)

In the manuscript entitled "A CSF-cytokine code predicts macrophage response polarity and tumor outcomes", authors reported that macrophage ontogeny determines whether cytokines promote or suppress tumor growth, using integrated transcriptomics, 3D tumor spheroids, and experimental metastasis models.

It is an interesting work, I have some questions:

- (1) Authors systematically mapped eight reference macrophage states. A question is how to determine such eight states, are there any other candidate states ?
- (2) Authors stated that the "ontogeny-cytokine code"—offers a predictive model of macrophage function. However, it is not clear what's the detailed predictive model ? A machine learning model or a statistical model ? And it is necessary to provide quantitative measurements like accuracy and AUC for evaluating the prediction power.
- (3) As shown in title, it is also necessary to validate the prediction of ontogeny-cytokine code on tumor outcomes by human cohort.
- (4) The cell-cell communication among these different states of macrophage should also be analyzed and provide more characterizations of macrophage response in TME.
- (5) It is better to have further discussion on the contribution of CSF-cytokine code in explaining the cancer treatment and diagnosis.

Version 1:

Reviewer comments:

Reviewer #1

(Remarks to the Author)

This is a revised version of the manuscript titled 'A CSF-cytokine code determines macrophage response polarity and tumor outcomes' by Schaer et al. Authors have addressed most comments and added key experiments, hence overall substantially improving the manuscript. A few concerns remain:

- One of the main issues is that some experiments were performed on GM-CSF macrophages while others were performed on M-CSF macrophages, and it's not always clear why this is the case. This clearly impacts the quality of the present manuscript. A stronger rationale should be provided to explain why the single cell analysis was performed only on GM-CSF macrophages: authors should more thoroughly explain why GM-CSF macrophages are more heterogeneous than M-CSF macrophages, which is really not clear from just the data presented in the manuscript, including Figure 1. The justification for performing a single cell analysis only on M-CSF spheroids in Figure 4 is also not clear, see 'These reproducible tumor cell response patterns enabled us to extrapolate macrophage-tumor cell relationships to the GM-CSF context' : this might be a risky assumption as data shown here actually support that M-CSF and GM-CSF macrophages might behave differently depending on cytokine treatment. Authors should revise this section.
- Figure 1: Authors mention Cav1 and Cspg4 but these genes are not visible in the figure.
- Figure 3B: If TGF- β is consistently pro-tumoral, why doesn't it increase ARG1 and SPP1 in both M-CSF and GM-CSF macrophages?
- Page 7: 'Together, these data validate... stimulatory capacity': this is a bit of an overstatement as only two proteins were investigated.
- Figure 3B – 5B-D – 6C-D: if 3 independent repeats of the same experiment were performed then maybe the figures should show an average of the data instead of representative data? This would more accurately represent data variability between biological replicates.
- Authors should further expand discussion on how they reconcile the fact that different macrophage states become more homogeneous at a transcriptomic level (Figure 4) when growing with cancer cells, while impacting tumor growth in vitro and in vivo differently (Figure 5, 6, 7).

Reviewer #2

(Remarks to the Author)

Authors have responded to all my concerned questions.

Version 2:

Reviewer comments:

Reviewer #1

(Remarks to the Author)

This is a revised version of the manuscript titled 'A CSF-cytokine code determines macrophage response polarity and tumor outcomes' by Schaer et al. Authors have addressed all the latest comments. It would be great if authors can make three more edits:

- Authors should more clearly state what additional data the single cell analysis in Figure 2 provides compared to the bulk RNA analysis in Figure 1: in its present form, the result section for Figure 2 reads too much like Figure 2 only validates results in Figure 1.

- Page 7 'To determine how our eight macrophage states influence cancer cells' = This should be rewritten as only M-CSF macrophages were tested here.

- Discussion: Authors should probably add the lack of single cell data on M-CSF macrophages and GM-CSF spheroids as limitations of the present manuscript.

Dear Editor and Reviewers,

We appreciate the constructive evaluation of our work. Below we provide a point-by-point response, followed by text changes that clarify methods, temper claims where appropriate, and improve figure legends. References to figures, pages, and methods correspond to the submitted manuscript.

We hope that our revised manuscript is acceptable for publication in *Communications Biology*.

Yours sincerely

Florence Vallelian

Reviewers' comments:

Reviewer #1 (Remarks to the Author):

This is an interesting study by Schaer et al., reporting data supporting that the tumor inhibitory vs supportive roles of macrophages are a function of both the developmental origin and the cytokines in the microenvironment of these cells. For example, here authors show that IL-4 induces tumor promoting functions in macrophages grown with M-CSF while it induces tumor inhibitory functions in macrophages grown with GM-CSF. Using bulk and single cell RNA-seq, authors investigated the transcriptome of the 8 macrophage states they generated through different combination of M-CSF or GM-CSF with various cytokines. Authors further validated the functions of each macrophage state through both in vitro co-culture (3D spheroid culture) and in vivo experiments. Overall this is a well written manuscript with valuable data. Yet, some of the data reported here still seem preliminary and sections are underdeveloped, including the single cell data analysis.

We appreciate the constructive evaluation of our work.

Extended validation of the functions of the macrophage states should be performed using other tumor cell lines.

We agree that validating our findings in additional tumor models is valuable. To address this, we have performed live-cell microscopy studies investigating the growth of mixed spheroids comprising macrophages and tumor cells using KP1.9 and TC-1 lung cancer cells. These experiments reproduced the ontogeny-dependent reversal of IL-4 effects observed with MC38 cells. Data are presented in Supplementary Figure 5. Given the

consistent results across three independent tumor lines, we believe this provides sufficient validation within the scope of the current study.

Some experiments were performed with M-CSF macrophages, others with GM-CSF macrophages: authors should keep experiments consistent throughout.

We aimed to provide a systematic approach comparing the phenotype and functions of M-CSF and GM-CSF macrophages. Therefore, we appreciate the opportunity to provide a more precise explanation of the rationale for the experiments presented in Figures 2 and 4. Figure 2 focuses on GM-CSF macrophages because this lineage is intrinsically more heterogeneous, warranting single-cell resolution. In contrast, M-CSF macrophages form a homogeneous population, for which bulk RNA-seq efficiently captures the relevant transcriptional differences. We highlight this distinction in the revised manuscript.

For the time-resolved mixed spheroid experiments presented in Figure 4, we used M-CSF macrophages to follow tumor-education dynamics over days 1→5. We focused the single-cell mapping on M-CSF spheroids, where our prior work has shown reproducible coupling between macrophage state and tumor-cell programs (PMID 38060331), and instead evaluated the GM-CSF arm with orthogonal functional readouts—spheroid growth, invasion, and in-vivo metastasis—that consistently recapitulated the ontogeny–cytokine code (including the IL-4 polarity inversion) across MC38, KP1.9, and TC-1 models. Polarizing cytokines were washed off before co-culture, so these outcomes reflect stable macrophage imprinting together with emergent tumor–macrophage cross-talk rather than continued cytokine exposure. Based on these reproducible findings and the high cost and complexity of scRNA-seq, we did not repeat this study with GM-CSF macrophages. We now clarify this rationale explicitly in the Results section of the revised manuscript.

In addition, further work on the mechanisms underlying the response of macrophages to the different CSF/cytokine combinations would be needed.

We agree that elucidating the underlying mechanisms is an essential next step, but a complete mechanistic dissection lies beyond the scope of the present study. We have now expanded the Limitations paragraph in the Discussion section, highlighting potential future investigation more explicitly, including ontogeny-specific chromatin landscapes and macrophage immunometabolism, which may account for the distinct responses to CSF/cytokine combinations.

Authors should also clarify the potential clinical application / impact of their data. See details below:

We added a paragraph to the Discussion referring to how to use the ontogeny context to guide cytokine modulation.

- Overall the results section should be expanded with further and more precise description of the figures. Reference to figures is also not always clear.

We have substantially revised the Results section to provide more detailed descriptions of each figure, including explicit references to individual panels, clarification of visual elements (axes, colors, spatial organization). The figure legends now include exact sample sizes (n), statistical tests with all pairwise comparisons, and clear definitions of "representative of three experiments" (independent biological replicates performed on different days with separate cell preparations).

- Figure 1: 'IL-10 shifted modestly toward the IFN-low quadrant' = this is not clear in Figure 1E. Also Ifi44 doesn't seem to be in any of the panels of Figure 1. GSEA and TF data are hardly discussed.

We re-phrased to avoid quadrant language and now state: "*IL-10 occupies an intermediate position along PC2 between the IFN- γ and TGF- β poles*". We removed the specific Ifi44 mention from text unless appearing in the displayed loading list; instead we now refer to "IFN-stimulated genes". We refined the GSEA and TF results.

- Figure 2: IL-4 GM-CSF macrophages are in fact split in 2 clusters, with only one of these 2 clusters enriched for NF-kB controlled gene expression. 'TGF-B and IL-10 co-clustered...inflammatory pole' (p.6): this seems to be true for TGF-B but not for IL-10 cells.

Principal Component analysis captures the dominant source of variation among the macrophage populations and reflects a global shift in their transcriptional programs. Therefore, movement along PC1 does not correspond to changes in a subset of genes or a specific cluster but rather represents a coordinated transcriptional reprogramming of the entire macrophage population, shifting from an inflammatory toward an anti-inflammatory or immunosuppressive state.

In Figure 2, we observed that IL-4-treated GM-CSF macrophages segregate into two transcriptionally distinct clusters, with only one of them enriched for NF-kB-regulated gene expression. Regarding the reviewer's comment on TGF- β and IL-10 macrophages, we agree that TGF- β -treated cells co-cluster toward the anti-inflammatory pole, while IL-10-treated macrophages occupy an intermediate position between the inflammatory and anti-inflammatory axes. We have revised the text accordingly.

Violin plots showing expression of specific genes in each cell population would help here. Was single cell RNA-seq also performed on M-CSF macrophages? In addition,

observations made from single cell RNA-seq data should be validated (expression of specific gene marker for instance).

The single-cell experiment presented in Figure 2 was performed exclusively with GM-CSF macrophages to resolve their intrinsic heterogeneity (see response to consistency comment above). We have now added supplementary violin plots displaying expression of the top PC1 loading genes (*Ccl22*, *Ccr7*) and PC2 loading genes (*C1qb*, *Ly6a*) across all treatment conditions (Supplementary Figure 2).

Additionally, the expression patterns of these marker genes are entirely consistent with bulk RNA sequencing data (n = 5 biological replicates per condition), which independently confirm the cytokine-specific transcriptional programs identified by single-cell analysis. Representative bulk RNA-seq expression profiles for key markers are shown in Supplementary Figure 2.

- Figure 3: 'consistent with previous observations' (p.6) = can you please clarify here?

We now specify that our finding (IL-10 and to a lesser extent TGF- β induce SPP1 expression selectively in GM-CSF-differentiated macrophages) is consistent with the results of Matsubara et al. (PMID: 36139536), who demonstrated that SPP1 is predominantly produced by GM-CSF-polarized macrophages in human lung adenocarcinoma, where TAM-derived SPP1 correlates with poor prognosis and chemoresistance. We have clarified this context in the revised text.

Figure 3E and F= these figures are hard to read, quantification, labels and stats are missing. Why weren't these experiments performed on M-CSF cells?

Figure 3E shows representative flow cytometry profiles, while CFSE dilution data were quantified and are presented in Figure 3F, with corresponding statistical analyses and n provided in the figure legend.

Expression of MHC-II in M-CSF-derived macrophages is extremely low. Accordingly, no antigen-specific CD4⁺ T-cell proliferation was observed, explaining why this experiment was not performed in the M-CSF condition.

- Figure 4-5: what's the rationale for performing these experiments at day 1 and 5 of coculture?

The two time points were chosen to capture distinct phases of macrophage-tumor cell interaction. Day 1 reflects the period when macrophages still retain their initial cytokine-induced phenotypes, allowing us to profile their early transcriptional states. Day 5 captures the emergence of tumor-cell phenotypes resulting from macrophage-mediated

reprogramming within the spheroid microenvironment. These time points were selected based on our previous single-cell RNA-seq analyses of mixed spheroids, which showed that TC identity diverges at later stages (PMID 38060331)..

- Figure 4: Were similar experiments performed using GM-CSF macrophages? + 'Importantly,... rather than ongoing cytokine signaling' = authors should consider what tumor cells and macrophages will produce and secrete in culture.

Please refer to my second comment "For the time-resolved mixed spheroid experiments (Figure 4), we used M-CSF macrophages to follow tumor-education dynamics over days 1→5...

We agree that paracrine signaling between tumor cells and macrophages is likely to influence the observed phenotypes. In our experimental design, polarizing cytokines were washed off before spheroid assembly, so any subsequent effects arise from factors expressed by tumor cells and macrophages during co-culture. We have now added a clarifying sentence in the Results emphasizing that our findings reflect the combined effects of stable macrophage programming and emergent tumor-macrophage cross-talk.

Figure 4E: it's not clear from these data what happens to the macrophages in culture, how different are they from day 1, any interesting differently genes to show here, or gene enrichment analysis between day 1 and day 5? From what data show here it seems that all macrophages, regardless of their original phenotype, adopt the same phenotype when in culture with cancer cells. It's not clear how the authors reach the following conclusion 'Notably, the apparent persistence... outcomes' re IFN- γ conditions. Overall this experiment would need to be repeated with GM-CSF macrophages and different tumor cell lines. Authors should also investigate specific markers to validate their single cell RNA-seq data.

To address this critical question, we performed a differential gene expression analysis comparing macrophages at day 1 and day 5 of co-culture. The top 100 up- and down-regulated genes were subjected to GSEA using the Hallmark and transcription factor target gene sets. This analysis revealed that day 1 macrophages displayed a more inflammatory transcriptional profile, whereas day 5 macrophages acquired an anti-oxidative, NRF2-driven phenotype, characterised by enrichment of genes involved in heme metabolism. Notably, the enrichment of NRF2 target genes at day 5 indicates a shift toward an immunosuppressive transcriptional state. This NRF2-dominated transcriptional state at day 5 is consistent with our previous single-cell analyses of mixed spheroids, where we identified NRF2-stress TAMs as a dominant tumor-educated macrophage phenotype that supports tumor growth and immunotherapy resistance (PMIDs 41176316, 38060331). The convergence of initially distinct macrophage states toward this shared NRF2+ profile demonstrates that tumor-derived signals progressively override initial cytokine-imprinted programs, regardless of ex vivo polarization state. We have added this analysis and

interpretation to the revised manuscript (Results, Figure 4 section, and Supplementary Figure 3).

- Figure 5 = authors should validate their comments on hypoxic stress

In multicellular tumor spheroid models, hypoxia develops gradually as spheroids grow in size and compactness due to limits in oxygen diffusion—so structural density alone can drive hypoxic gradients (PMID: 32357910).

KP1.9 cancer cells should also be used in co-culture with M-CSF cells.

This experiment was already performed. Co-cultures of KP1.9 cancer cells with M-CSF macrophages are included in the revised manuscript and demonstrate the same ontogeny-dependent reversal of IL-4 effects observed with MC38 cells (Supplementary Figure 3).

Figure 7D–E: It's not clear what the point of these panels is. Authors should clarify the potential clinical impact of these data. In the discussion, authors state 'Our data argue against global TAM depletion,' which is true but does not come across clearly enough in the manuscript. Further experiments aimed at supporting this statement could be performed. Authors should further clarify how their data could help patient treatment and care, and for which cancers specifically.

We thank the reviewer for this helpful comment. Figure 7D presents z-score heatmaps that standardize diverse functional readouts onto a common scale, enabling direct comparison of cytokine effects within each ontogeny. Figure 7E presents differential z-scores (M-CSF minus GM-CSF) that quantitatively identify which cytokines exhibit ontogeny-dependent switching (IL-4, IL-10) versus ontogeny-independent effects (IFN- γ , TGF- β). Together, these panels establish a "CSF-cytokine code" that categorizes macrophage functional responses and provides a framework for therapy design.

We have substantially revised the Results section (Figure 7 description) and Discussion to clarify this synthesis and its implications. Specifically, we now explicitly state that global TAM depletion strategies would eliminate both anti-tumoral (e.g., IFN- γ - or IL-4-stimulated GM-CSF macrophages) and pro-tumoral macrophages, potentially compromising anti-tumor immunity. Instead, CSF-stratified cytokine modulation offers precision targeting: (1) In M-CSF-dominant tumors, blocking IL-4 or IL-10 signaling could selectively deplete ARG1+ immunosuppressive TAMs while preserving IFN- γ -responsive populations; (2) In GM-CSF-enriched tumors, blocking IL-10/TGF- β while delivering IL-4 could enhance antigen presentation and T-cell priming; (3) Combined CSF1R inhibition with GM-CSF+IL-4 administration could simultaneously deplete M-CSF-dependent protumoral TAMs and expand GM-CSF-dependent antigen-presenting macrophages. These strategies are now

detailed in a new paragraph in the Discussion (Clinical Implications section), with reference to recent clinical evidence that IL-4 pathway modulation can be therapeutically exploited

- Figure 3B – 5B-D – 6C-D = it's not clear what these graphs show. 'representative of three independent experiments' = can you please clarify what this means? Were any statistical tests run on these experiments?

Figure 3B shows time-course data where each curve represents the mean integrated fluorescence intensity from six imaging fields per well. This experiment was performed three times with separate bone marrow preparations, yielding consistent results; one representative experiment is shown.

Figures 5B–D and 6C–D display quantitative kinetic measurements: 5B/D show tumor spheroid growth (integrated GFP fluorescence over 10 days), and 6C/D show matrix invasion (invading cell area over 72 hours). For each time-course panel, curves represent mean \pm SE of technical replicates (8-10 for growth, 3-4 for invasion) from one experiment. The middle panels show area under the curve (AUC) quantification with statistical analysis (one-way ANOVA with Dunnett or Tukey-Kramer post-hoc tests), exact n values, and individual data points displayed as open circles. The right panels show representative brightfield images at multiple timepoints illustrating the quantified phenotypes.

"Representative of three independent experiments" means that the entire protocol—from bone marrow isolation through macrophage differentiation, spheroid formation, and imaging—was performed three separate times on different days, and all replicates produced qualitatively similar results with consistent statistical outcomes. We present data from one experiment to avoid pseudoreplication in statistical analyses while confirming reproducibility across biological replicates. We have clarified these details in the revised figure legends.

Reviewer #2 (Remarks to the Author):

In the manuscript entitled “A CSF-cytokine code predicts macrophage response polarity and tumor outcomes”, authors reported that macrophage ontogeny determines whether cytokines promote or suppress tumor growth, using integrated transcriptomics, 3D tumor spheroids, and experimental metastasis models.

It is an interesting work, I have some questions:

We thank the reviewer for the positive assessment of our work and for the thoughtful questions that helped us clarify and strengthen the manuscript.

(1) Authors systematically mapped eight reference macrophage states. A question is how to determine such eight states, are there any other candidate states ?

We selected canonical macrophage states based on two widely accepted differentiation axes: ontogeny (M-CSF vs. GM-CSF) and four prototypical cytokines (IFN- γ , IL-4, IL-10, and TGF- β) that represent the main functional polarization cues described in the literature. This 2 \times 4 matrix provides a comprehensive and experimentally tractable framework to capture the principal macrophage activation modes relevant to tumor biology. We fully acknowledge that additional states exist in vivo, driven by other mediators such as prostaglandins, adenosine, or metabolic signals; however, inclusion of these would have compromised the systematic comparability of the present study. These additional activation contexts are mentioned in the Discussion as potential extensions of the ontogeny–cytokine framework.

(2) Authors stated that the "ontogeny-cytokine code"—offers a predictive model of macrophage function. However, it is not clear what's the detailed predictive model? A machine learning model or a statistical model? And it is necessary to provide quantitative measurements like accuracy and AUC for evaluating the prediction power.

We thank the reviewer for this clarification. By "ontogeny-cytokine code," we refer to a conceptual framework that categorizes macrophage functional responses based on the combination of developmental origin (M-CSF vs. GM-CSF) and cytokine environment (IFN- γ , IL-4, IL-10, TGF- β). This framework is not a machine learning or statistical prediction model requiring AUC/accuracy metrics, but rather a biological classification system derived from systematic experimental measurements across multiple functional readouts (transcriptomics, growth, invasion, metastasis, T-cell activation).

The "code" enables prediction in the sense that: (1) Once a tumor's dominant CSF context is known (M-CSF vs. GM-CSF), the framework categorizes whether a given cytokine will promote or suppress tumor progression; (2) IL-4 and IL-10 exhibit ontogeny-dependent polarity switching (pro-tumoral in M-CSF, anti-tumoral in GM-CSF for IL-4; weak in M-CSF, strong pro-tumoral in GM-CSF for IL-10); (3) IFN- γ and TGF- β show ontogeny-independent effects (consistently anti- and pro-tumoral, respectively).

To avoid confusion with computational prediction models, we have revised the title to read: "A CSF-cytokine code determines macrophage response polarity and tumor outcomes." We have also clarified in the Discussion that this framework provides a biological basis for designing context-specific macrophage-targeted therapies rather than a quantitative prediction algorithm.

(3) As shown in title, it is also necessary to validate the prediction of ontogeny-cytokine code on tumor outcomes by human cohort.

We agree that clinical validation in human cohorts would strengthen the translational relevance of our findings. To avoid overstating our current evidence, we have revised the title from "predicts" to "determines," emphasizing that our framework establishes causal relationships in controlled experimental systems rather than claiming validated clinical prediction.

We note in the Discussion that validating the ontogeny-cytokine code in human tumors will require: (1) Spatially resolved profiling to distinguish M-CSF-dominant versus GM-CSF-enriched TAM populations within patient samples; (2) Correlation of CSF/cytokine signatures with clinical outcomes; and (3) Prospective studies testing whether CSF-stratified cytokine interventions improve therapeutic responses. Recent studies have begun to implement IL-4 pathway modulation in clinical trials (Crowley et al., 2025; Mollaoglu et al., 2024), laying the groundwork for testing our framework's predictions in patients. This important future direction is now explicitly stated in the Discussion.

(4) The cell-cell communication among these different states of macrophage should also be analyzed and provide more characterizations of macrophage response in TME.

We agree that analyzing cell–cell communication among macrophage states would provide additional insights into their roles within the tumor microenvironment. However, our current datasets were not designed to robustly infer macrophage–macrophage interactions, as they focus on macrophage–tumor cell co-cultures and transcriptional programs driving tumor outcomes. We therefore limited our analyses to experimentally supported macrophage–tumor cell interactions.

Dear Dr. Karlsson Rosenthal,

We appreciate the constructive evaluation of our work. Below, we provide a point-by-point response to the reviewers' concerns. References to figures, pages, and methods correspond to the submitted manuscript.

We hope that our revised manuscript is acceptable for publication in *Communications Biology*.

Yours sincerely

Florence Vallelian

Reviewer #1 (Remarks to the Author):

This is a revised version of the manuscript titled 'A CSF-cytokine code determines macrophage response polarity and tumor outcomes' by Schaer et al. Authors have addressed most comments and added key experiments, hence overall substantially improving the manuscript.

We thank the reviewer for their time and effort in evaluating our work and for their insightful comments, which have significantly strengthened the manuscript.

A few concerns remain:

- One of the main issues is that some experiments were performed on GM-CSF macrophages while others were performed on M-CSF macrophages, and it's not always clear why this is the case. This clearly impacts the quality of the present manuscript. A stronger rationale should be provided to explain why the single cell analysis was performed only on GM-CSF macrophages: authors should more thoroughly explain why GM-CSF macrophages are more heterogenous than M-CSF macrophages, which is really not clear from just the data presented in the manuscript, including Figure 1. The justification for performing a single cell analysis only on M-CSF spheroids in Figure 4 is also not clear, see 'These reproducible tumor cell response patterns enabled us to extrapolate macrophage-tumor cell relationships to the GM-CSF context' : this might be a risky assumption as data shown here actually support that M-CSF and GM-CSF macrophages might behave differently depending on cytokine treatment. Authors should revise this section.

Regarding the heterogeneity of GM-CSF macrophages (Figure 2): Our decision to perform single-cell analysis specifically on the GM-CSF lineage is grounded in established immunology. As described by Helft et al. and Vallelian et al. (Refs. 17 and 18), GM-CSF bone marrow cultures are intrinsically heterogeneous, generating both macrophages and a distinct population of Ccl22+ Ccr7+ dendritic cell-like inflammatory macrophages; under certain conditions, even neutrophils. M-CSF cultures, by contrast, yield a homogeneous macrophage population. While our bulk RNA-seq (Figure 1) effectively captures population-level shifts, it cannot resolve whether cytokines differentially affect these subpopulations. The scRNA-seq in Figure 2 was therefore essential to demonstrate that IL-4 drives a specific inflammatory trajectory in the "DC-like" subset of the GM-CSF lineage—a nuance that does not exist in the M-CSF lineage and would be lost in bulk analysis. We have revised the text in the Results section (Page 5) to explicitly cite this established heterogeneity as the driver for this experimental choice. We also emphasize that our principal transcriptome-based conclusions align with the bulk RNA-seq studies shown in Figure 1, which encompass all lineage x cytokine combinations (Macrophage ontogeny and cytokine environment create distinct transcriptional landscapes). To explain GM-CSF macrophage heterogeneity in more detail here, we include a Figure from Vallelian, F. et al. (Ref. 21) that, using scRNA-seq, compares a homogeneous M-CSF macrophage population with heterogeneous GM-CSF macrophage cultures under control and heme-stressed conditions.

B UMAP plot showing cells colored by cell type.
C Factorial UMAP plots of the dataset separated by culture supplement (M-CSF versus GM-CSF) and treatment condition (vehicle versus heme exposure). The data points are colored by cell type (orange = macrophages, purple = neutrophils, green = DC-like).

[from Vallelian, F. et al. Heme-stress activated NRF2 skews fate trajectories of bone marrow cells from dendritic cells towards red pulp-like macrophages in hemolytic anemia. *Cell Death Differ* 29, 1450–1465 (2022).]

dendritic cell-like inflammatory macrophages (Helft et al. 2015; Vallelian et al. 2022). To account for this heterogeneity,...

Regarding the M-CSF focus in spheroids (Figure 4): We selected M-CSF macrophages for the time-resolved scRNA-seq analysis (Figure 4) to investigate "tumor education" and the convergence toward NRF2-stress states, which we previously characterized in M-CSF models (Refs. 7, 22). While we agree that M-CSF and GM-CSF macrophages have different starting states (as shown in Figure 1), our functional data in Figures 5, 6, and 7 demonstrate that the outcomes are distinct and predictable based on the ontogeny-cytokine code. Given that the functional "inversion" of IL-4 was robustly validated across three tumor cell lines (MC38, KP1.9, TC-1) and in vivo models using GM-CSF macrophages, we reasoned that repeating the resource-intensive scRNA-seq time course for the GM-CSF lineage would yield diminishing returns. We have revised the text to remove the phrase "extrapolate... relationships" to avoid overstatement.

We have added the following statement to the respective results section on page 7:

We selected M-CSF macrophages for this analysis because their homogeneous starting population enables cleaner resolution of tumor-induced transcriptional changes, and because our prior single-cell work with mixed spheroids showed that tumor-cell transcriptional patterns closely mirror macrophage functional phenotypes (tumoricidal vs. non-tumoricidal).^{7,22} While parallel scRNA-seq of GM-CSF spheroids would provide additional resolution, our functional data (Figures 5–7) directly demonstrate that ontogeny-dependent outcomes are robust and reproducible across both lineages.

- Figure 1: Authors mention Cav1 and Cspg4 but these genes are not visible in the figure.

We thank the reviewer for spotting this inconsistency. We have updated the text to refer to **Mmp14, Pmepa1, and Notch4**, which are representative of the signatures displayed in Figure 1.

- Figure 3B: If TGF- β is consistently pro-tumoral, why doesn't it increase ARG1 and SPP1 in both M-CSF and GM-CSF macrophages?

This is an important point that underscores the complexity of macrophage polarization. Our data indicate that while ARG1 and SPP1 are strong markers of specific pro-tumoral states (e.g., IL-4-driven M-CSF), they do not represent a universal "pro-tumoral" signature. Consequently, the absence of these two specific markers does not preclude pro-tumoral function. Notably, TGF- β promoted protumoral behaviors without inducing ARG1 or SPP1,

consistent with a distinct SMAD-driven matrix-remodeling program (e.g., Mmp14/Pmepa1/EMT hallmarks; Fig. 1 and Figs 6–7).

- Page 7: 'Together, these data validate... stimulatory capacity': this is a bit of an overstatement as only two proteins were investigated.

We agree and have modified the sentence to read: "Together, these data support..."

- Figure 3B – 5B-D – 6C-D: if 3 independent repeats of the same experiment were performed then maybe the figures should show an average of the data instead of representative data? This would more accurately represent data variability between biological replicates.

We acknowledge the reviewer's preference for pooled data. However, in long-term live-cell imaging (Incucyte) assays spanning several days, we observe run-to-run scaling differences in raw fluorescence units (RFU) between independent experiments performed weeks apart (due to slight differences in fluorophore maturation, baseline autofluorescence, or calibration). Furthermore, subtle differences in experimental conditions across runs can lead to shifts in growth trajectories. Pooling these raw curves introduces high variance that obscures the highly reproducible kinetic patterns observed within every individual experiment. To demonstrate this robustness without creating misleading error bars, we present a representative experiment (where N=8-10 replicate spheroids provide statistical power for that run). To address the reviewer's concern regarding reproducibility, we have included plots from independent replicate experiments below, demonstrating that the relative differences between treatment groups are consistent across repeats.

Furthermore, we added a statement to the respective method section on page X

Due to technical batch effects, data are presented as representative runs with intra-experiment replicates (n=8–10).

Replication-Figure 3B

Replication-Figure 5B

Replication-Figure 5D

Replication-Figure 6B

Replication-Figure 6D

- Authors should further expand discussion on how they reconcile the fact that different macrophage states become more homogenous at a transcriptomic level (Figure 4) when growing with cancer cells, while impacting tumor growth in vitro and in vivo differently (Figure 5, 6, 7).

We have expanded the Discussion to address this apparent paradox. Our data suggest that the "transcriptional convergence" observed by Day 5 in Figure 4 reflects a dominant adaptation to the spheroid microenvironment (specifically, an NRF2-mediated oxidative stress response) that overlays the initial polarization state. However, the functional outcome (spheroid growth vs. killing) is likely determined by the initial macrophage state (Day 0–3) and the immediate interactions following co-culture, before this convergence is fully established. We have added the following statement to the Discussion:

Page 13: Our data suggest that while tumor-derived environmental signals eventually drive macrophages toward a convergent transcriptional state in spheroids, the initial cytokine-imprinted functionality determines the trajectory of tumor growth or suppression. This implies that the ontogeny-cytokine code acts as a critical gatekeeper during the early phases of tumor-macrophage interaction, setting a functional course that persists even as transcriptomic profiles begin to align under environmental pressure.

Reviewer #2 (Remarks to the Author):

Authors have responded to all my concerned questions.

We thank the reviewer for the time and effort spent evaluating and improving our work.

Dear Editor and Reviewers,

We appreciate the constructive evaluation of our work. Below we provide a point-by-point response, followed by text changes. We also edit our manuscript to comply with Communications Biology format requirements.

In addition, we submit a hand-painted artwork depicting mixed-cell spheroid invasion for consideration in the Featured Image section.

We hope that our revised manuscript is suitable for publication in *Communications Biology*.

Yours sincerely

Florence Vallelian

Reviewers' comments:

Reviewer #1 (Remarks to the Author):

This is a revised version of the manuscript titled 'A CSF-cytokine code determines macrophage response polarity and tumor outcomes' by Schaer et al. Authors have addressed all the latest comments.

It would be great if authors can make three more edits:

- Authors should more clearly state what additional data the single cell analysis in Figure 2 provides compared to the bulk RNA analysis in Figure 1: in its present form, the result section for Figure 2 reads too much like Figure 2 only validates results in Figure 1.

We agree that the added value of the single-cell analysis should be explicitly articulated. We have revised the opening paragraph of the Figure 2 Results section to clearly state that scRNA-seq was performed to determine whether the cytokine-driven transcriptional axes identified in bulk RNA-seq reflect population-wide shifts or are driven by rare subpopulations within heterogeneous GM-CSF cultures.

Compared to M-CSF macrophages, GM-CSF macrophage cultures are intrinsically heterogeneous, generating both macrophages and a distinct population of MHCII+ Ccl22+ Ccr7+ dendritic cell-like inflammatory macrophages.^{17,18} To determine whether the transcriptional axes identified by bulk

RNA-seq reflect uniform population-wide shifts or arise from rare subpopulations within GM-CSF cultures, we performed multiplexed scRNA-seq on 49,441 GM-CSF–derived macrophages across all cytokine treatment conditions. UMAP of the combined dataset revealed treatment-specific clustering patterns (**Figure 2A**).

- Page 7 'To determine how our eight macrophage states influence cancer cells' = This should be rewritten as only M-CSF macrophages were tested here.

We thank the reviewer for pointing this out. The text has been revised to explicitly state that the time-resolved single-cell spheroid analysis was performed with M-CSF-derived macrophages only.

The revised sentence now reads:

“To determine how M-CSF-macrophage states influence cancer cells, we conducted time-resolved, multiplexed scRNA-seq of mixed 3D spheroids containing M-CSF macrophages and MC38 colon carcinoma cancer cells (**Figure 4A**)”.

This revision ensures that no overgeneralization to all eight macrophage states is implied.

- Discussion: Authors should probably add the lack of single cell data on M-CSF macrophages and GM-CSF spheroids as limitations of the present manuscript.

We agree and have now added an explicit statement in the Discussion addressing this point.

The new text appears in the Discussion section under “Several limitations warrant consideration.”

Several limitations warrant consideration. Our reductionist approach excludes key tumor microenvironment components (vasculature, hypoxia, adaptive immunity) and tests only four cytokines, excluding prostaglandins, adenosine, heme, and other tumor metabolites known to modulate TAMs.^{38,39} However, this simplified system was essential for establishing ontogeny-cytokine interactions without confounding variables. Finally, we limited time-resolved single-cell analysis in spheroids to M-CSF macrophages. Although GM-CSF macrophage dynamics were not profiled at single-cell resolution in 3D culture, the ontogeny-dependent effects on

tumor-cell growth, invasion, and metastasis were reproducible across independent functional assays and consistent with the initial cytokine imprinting of each lineage.

In addition to the bulk RNA-seq analyses presented in Figure 1, we have independently performed single-cell RNA-sequencing of M-CSF–derived macrophages in the past. These analyses did not reveal discrete transcriptional subpopulations, supporting the homogeneity of M-CSF cultures. Based on these data, we do not consider the absence of single-cell profiling in the current manuscript to represent a limitation, as additional single-cell analyses would be unlikely to alter the conclusions of the study.